# On a Magical Mystery Tour with 8-Bromo-Cyclic ADP-Ribose: From All-or-None Block to Nanojunctions and the Cell-Wide Web

**DOI:** 10.3390/molecules25204768

**Published:** 2020-10-16

**Authors:** A. Mark Evans

**Affiliations:** Centre for Discovery Brain Sciences and Cardiovascular Science, Edinburgh Medical School, Hugh Robson Building, University of Edinburgh, Edinburgh EH8 9XD, UK; mark.evans@ed.ac.uk

**Keywords:** 8-bromo-cADPR, cADPR, calcium, hypoxic pulmonary vasoconstriction, sarcoplasmic reticulum, K_V_7, ryanodine receptor, two pore channel 2, arterial smooth muscle, nanojunction, nanospace, nanocourse, nucleus, nuclear invagination

## Abstract

A plethora of cellular functions are controlled by calcium signals, that are greatly coordinated by calcium release from intracellular stores, the principal component of which is the sarco/endooplasmic reticulum (S/ER). In 1997 it was generally accepted that activation of various G protein-coupled receptors facilitated inositol-1,4,5-trisphosphate (IP_3_) production, activation of IP_3_ receptors and thus calcium release from S/ER. Adding to this, it was evident that S/ER resident ryanodine receptors (RyRs) could support two opposing cellular functions by delivering either highly localised calcium signals, such as calcium sparks, or by carrying propagating, global calcium waves. Coincidentally, it was reported that RyRs in mammalian cardiac myocytes might be regulated by a novel calcium mobilising messenger, cyclic adenosine diphosphate-ribose (cADPR), that had recently been discovered by HC Lee in sea urchin eggs. A reputedly selective and competitive cADPR antagonist, 8-bromo-cADPR, had been developed and was made available to us. We used 8-bromo-cADPR to further explore our observation that S/ER calcium release via RyRs could mediate two opposing functions, namely pulmonary artery dilation and constriction, in a manner seemingly independent of IP_3_Rs or calcium influx pathways. Importantly, the work of others had shown that, unlike skeletal and cardiac muscles, smooth muscles might express all three RyR subtypes. If this were the case in our experimental system and cADPR played a role, then 8-bromo-cADPR would surely block one of the opposing RyR-dependent functions identified, or the other, but certainly not both. The latter seemingly implausible scenario was confirmed. How could this be, do cells hold multiple, segregated SR stores that incorporate different RyR subtypes in receipt of spatially segregated signals carried by cADPR? The pharmacological profile of 8-bromo-cADPR action supported not only this, but also indicated that intracellular calcium signals were delivered across intracellular junctions formed by the S/ER. Not just one, at least two. This article retraces the steps along this journey, from the curious pharmacological profile of 8-bromo-cADPR to the discovery of the cell-wide web, a diverse network of cytoplasmic nanocourses demarcated by S/ER nanojunctions, which direct site-specific calcium flux and may thus coordinate the full panoply of cellular processes.

## 1. Introduction

Cells select for one or a combination of distinct functions through calcium signalling. Therefore, stimuli must induce different calcium signals to select for one or other of a variety of specific cellular processes, such as cell activation, inhibition and proliferation, which additionally requires changes in gene expression [1]. A variety of highly specialised, intracellular ion channels evolved for this purpose [2,3]. However, despite the extraordinarily detailed mapping of the temporal characteristics of both unitary and macroscopic calcium signals across a variety of cell types [4,5,6,7,8,9,10], how cells deliver the diverse range of site- and function-specific calcium signals necessary to coordinate the full panoply of cellular processes remains enigmatic [11,12].

The primary intracellular calcium store is the sarco/endoplasmic reticulum (S/ER), which forms a contiguous organelle from its origin at the outer nuclear membrane to the periphery of all cells [13]. Calcium signals with clear diversities of form and function are delivered via the S/ER, in a manner that meets the specific functional requirements of a given cell type [6,7,10,12,14]. Irrespective of cell type the current consensus is that within a wide, open cytoplasm highly localised calcium signals (e.g., calcium sparks) direct region-specific functions, while propagating global calcium signals coordinate primary cell-specific functions (e.g., muscle contraction, secretion). Adding to this, adjustments to gene expression are presumed to be governed by variations in the spatiotemporal patterns of global calcium transients that gain unrestricted entry to the nucleoplasm across the nuclear envelope and its invaginations [15,16,17,18,19].

This review will retrace the steps in the development and testing of a novel hypothesis on the mechanism of intracellular calcium signalling, that was first prompted by the very peculiar pharmacology of 8-bromo-cyclic adenosine diphosphate-ribose (8-bromo-cADPR; Figure 1) [20]. Importantly, 8-bromo-cADPR had been proposed to be a competitive antagonist of cyclic adenosine diphosphate-ribose (cADPR), a novel calcium mobilising pyridine nucleotide [21]. The pharmacological profile of 8-bromo-cADPR of which I speak was at first sight surprising, but then became curiouser, and curiouser still. There seemed to be one explanation for outcomes and one explanation only, that different calcium signals must be delivered by different SR compartments, into functionally distinct cytoplasmic spaces, somehow demarcated by junctions formed by the SR [11,22,23,24,25]. Our further studies ultimately led to the discovery of the cell-wide web, a network of cytoplasmic nanocourses demarcated by S/ER, which direct calcium signals in a manner that may be sufficient to allow for stimulus-specified coordination of the full panoply of cellular processes.

However, the path to the cell-wide web did not begin with considerations on calcium signalling *per se*, but with the discovery of a novel potassium current in acutely isolated pulmonary arterial smooth muscle cells. Therefore, in order to understand the “logic” of each step along this experimental road, the story must begin here.

## 2. A Tale of Two Channels and Perhaps a Few Dollars More

During 1994 I was tasked with characterising potassium channels in porcine pulmonary arterial myocytes, with the aim of determining whether, as postulated, exposure to hypoxia during the perinatal period altered normal developmental changes in potassium currents (kinetics and pharmacology) and in doing so contributed to the development of persistent pulmonary hypertension of the newborn. It is satisfying, with hindsight, to report that this hypothesis was confirmed [26], which deserves mention because these studies were riddled with periodic delays and lengthy periods of out and out failure. That said, these delays allowed me the freedom for pure unadulterated exploration of an unknown.

It had been proposed that the resting membrane potential of pulmonary arterial myocytes was determined not by the large conductance voltage- and calcium-activated potassium current (BK_Ca_; KCa1.1 to the youth of today) but by a delayed rectifier potassium current (IK_V_), the primary component of which we now know is carried by K_V_1.5 [27,28,29,30]. At the time pharmacological tools were limited and poorly selective, while molecular interventions had yet to come of age. Therefore, I considered other ways in which I might test whether it was indeed the delayed rectifier that set the resting membrane potential of pulmonary arterial myocytes. Prolonged depolarisation was thought to evoke voltage- and time-dependent inactivation of all known K_V_ currents. Therefore, I designed a protocol that would remove these potassium channels through voltage-dependent inactivation. I held cells under voltage-clamp for 1–2 h at 0 mV and then switched immediately to current clamp (I = 0), leaving each cell to report its residual resting membrane potential (experimental protocol reported in [31]). In those few experiments that lasted the course, I was astonished to find that the resting membrane potential remained the same as it was before the inactivation protocol had been engaged, not altered by even a single mV. This led to the discovery of a novel low threshold voltage-gated, low conductance potassium current that was activate at the resting membrane potential of pulmonary arterial myocytes (approximately −60 mV), which I named IK_N_ because it did not inactivate [32]. By contrast, the threshold for activation of IK_V_/K_V_1.5 is −40 mV, +20 mV above the resting membrane potential recorded in acutely isolated cells with a seal resistance ≥10 GΩ [32], or for that matter in smooth muscle cells in-situ in an intact artery [33]. The characteristics of IK_N_ were so strikingly similar to the neuronal M-current [32] that subsequent studies of others inevitably demonstrated, as predicted [32], that they were both conferred by the same potassium channel type, namely K_V_7 [34,35]. At this point I was asked to involve a fellow post-doctoral researcher, Oleg Osipenko, in my further investigations on IK_N_, which confirmed my findings, and furthermore suggested that IK_N_ might be inhibited during hypoxia [31]. It was therefore proposed that inhibition of IK_N_ rather than IK_V_ [27,28,29,36] was most critical to the induction of hypoxic pulmonary vasoconstriction (HPV), a reflex response of pulmonary arteries to falls in airway and alveolar oxygen availability [37]. It was made clear to me that the effects of hypoxia on IK_N_ were not for me to test, and to this day I have not done so. Rather than any long-term compliance on my part, this was due to a twist in the tale that I was presented with on setting up my own laboratory to further my independent studies on IK_N_.

## 3. K_N_ or Not K_N_ That Was the Question

HPV is a local response mediated by mechanisms intrinsic to the smooth muscles and endothelia of the pulmonary blood vessels, that helps optimise ventilation-perfusion matching by diverting blood flow through the path of least resistance, from oxygen-deprived to oxygen-rich areas of the lung; by contrast most aspects of the systemic vasculature dilate during hypoxia to ensure that oxygen availability meets the needs of the organs and tissues thus supplied [38]. HPV is triggered by airway and alveolar hypoxia [37,39], rather than by vascular hypoxaemia [40], through the constriction of pre-capillary resistance arteries within the pulmonary circulation, in a manner coordinated by signalling pathways that are intrinsic to their smooth muscles and endothelial cells [41,42,43,44], independently of blood-borne mediators or the autonomic nervous system [45,46]. The initiation phase of acute HPV is primarily driven by smooth muscle constriction [41], with a threshold *P*O_2_ ≈ 80 mmHg [24]. The cell-specific expression of atypical nuclear encoded subunits of the mitochondrial electron transport chain, such as COX4I2, may confer the acute sensitivity of these oxygen-sensing cells to physiological hypoxia [47,48], while increased expression, relative to systemic arteries, and activation of the AMP-activated protein kinase appears critical to induction of HPV downstream of mitochondria [49,50], including therein direct phosphorylation and inhibition of K_V_1.5 channels that underpin the aforementioned IK_V_ [50,51]. That is as it stands now, but we knew little of this back then.

When evoked in isolated pulmonary arterial rings by switching from normoxic to hypoxic gas mixtures, HPV presents as a biphasic response (Figure 2), with an initial transient constriction (phase 1) being followed by a slow tonic constriction (phase 2) [41,52,53,54]. Both phases of constriction are superimposed on each other; that is to say, they are discrete events that are initiated concomitantly upon exposure of pulmonary arteries to hypoxia. The initial transient constriction peaks within 5–10 min of the onset of hypoxia, whilst the underlying, tonic constriction peaks after a further 30–40 min of hypoxia. The gradual amplification of phase 2 is driven by the release of an endothelium-derived vasoconstrictor [43,55,56], the identity of which is still open to question. Therefore, after the endothelium is removed the phase 1 constriction declines to a maintained plateau [41].

The prevailing hypothesis back then was, and still now remains so in the eyes of some, that HPV is triggered by the inhibition of voltage-gated potassium channels, consequent membrane depolarisation and voltage-gated calcium influx [27,28,29,36,47,57,58].

In 1996 I set about designing a series of experiments that would confirm the preeminent role of potassium channels in HPV, and IK_N_ in particular, using experimental protocols that extended my work on acutely isolated smooth muscles to isolated pulmonary arterial rings and the whole rat lung in-situ, with the kind support of Dr Piers Nye. In these investigations I was ably assisted by my first PhD student, Michelle Dipp, who consistently obtained acute and robust HPV using isolated pulmonary artery rings without the need for the application of pre-tone by way of an applied vasoconstrictor; a common practice in other labs, perhaps to overcome less nimble dissection, and/or the use of anaesthetics during sacrifice, such as pentobarbitone (see also [59]).

My first priority was to complete the most obvious and critical experiment the literature appeared to lack. In my mind this was a sure-fire bet, IK_N_ would win the day. We (Michelle) would remove extracellular calcium from the medium using the calcium chelator EGTA, and balance the loss of this divalent cation by addition of equimolar quantities of magnesium. Constriction of pulmonary artery rings during hypoxia would be abolished, there could be no doubt about that, and then we would move on. To be doubly sure, I suggested we control for any input from the endothelium by removing it from paired artery rings. Michelle and I met at the end of the first week to discuss her preliminary findings. To my astonishment, she informed me that there was only a very little change in the response to hypoxia of the endothelium-denuded artery [41]. I think I said something along the lines of, “What, show me the records!”. She did and sure enough HPV (phases 1 and 2) was at best partially inhibited, whether one considered each individual experiment, or the average. Michelle replied “Is that not what you expected?”; she was not surprised at all as she had simply been asked to tell me what she found. Moreover, the constriction induced through membrane depolaristion in response to high extracellular potassium (80 mM; negative control) was abolished as expected, confirming the most unexpected of outcomes, that neither potassium channel inhibition or voltage-gated calcium influx was a pre-requisite for acute HPV. Michelle offered me solace by revealing records from intact arteries, where the endothelium-dependent component of HPV had been attenuated, suggesting that calcium influx into the endothelium was required for vasoconstrictor release. This did not help my mindset at the time, as I was so focused on the mechanism at play within the smooth muscles. However, intrinsic HPV was and, as attested to by removal of the endothelium, it was most likely triggered by calcium release from intracellular stores within pulmonary arterial myocytes. This was confirmed by pre-incubation of pulmonary arteries with ryanodine and caffeine, which blocked calcium release from the SR via RyRs without depleting the SR calcium store. Phase 1 and phase 2 of HPV were duly abolished, but this intervention did not affect constriction by high potassium [41]. Perhaps most astonishingly, when considered alongside the fact that HPV could be induced in the absence of extracellular calcium, this outcome indicated that calcium released from the SR was somehow locked inside the cell, because HPV was sustained for a long time in calcium-free media, longer than the limits of any test we carried out (i.e., hours). It should however be noted that the maintained phase of constriction of pulmonary artery rings without endothelium was attenuated by up to 50% in calcium-free medium, consistent with the view that HPV is supported by consequent activation of a store-depletion-activated calcium entry pathway(s) [60].

Several investigations had previously suggested that HPV may be facilitated in part by calcium release from SR stores [52,61,62,63,64], but none had so clearly demonstrated that this was initiated independent voltage-gated calcium influx and endothelium-derived vasoconstrictors. Hence general acceptance of the dominant hypothesis of the time, that HPV was triggered by K_V_ channel inhibition and consequent voltage-gated calcium influx [27,28,29,36,47,57,58], which had now been debunked, from my perspective at least.

## 4. Two for Tea: Another Awkward Moment in Time

My first post-doctoral researcher, Francois Boittin, joined us the following year. I set Francois a parallel study, to examine the cellular mechanism by which a β-adrenoceptor agonist, isoprenaline, mediated relaxation of pulmonary arterial smooth muscles. It turned out that while Francois was an excellent electrophysiologist, he was not adept at small or large artery dissection; when supplied with Francois’ arteries, Michelle pointed out that they were almost dead on arrival, “Mark, they may contract to potassium, just, but not hypoxia!”. In short, this study became a team effort, naturally.

Like many other Gs coupled receptors, β-adrenoceptors had been shown to activate adenylyl cyclase, increase cyclic adenosine monophosphate (cAMP) levels and mediate vasodilation in a protein kinase A (PKA)-dependent manner. In a variety of smooth muscles PKA appeared to facilitate relaxation by: (1) increasing SR filling through SERCA pump acceleration, consequent to PKA-dependent phosphorylation of the SERCA inhibitory protein phospholamban, which promotes its dissociation from SERCA [65,66]; (2) increasing calcium spark frequency through peripheral RyRs due to increased SR calcium load [67,68]; (3) facilitating activation of plasmalemma resident voltage- and calcium-activated large conductance potassium channels (BK_Ca_, KCa1.1) and thus membrane hyperpolarisation, directly through BK_Ca_ channel phosphorylation [69] and indirectly through increased calcium spark frequency from regions of the SR proximal to the plasmalemma [8,70]. Thereafter, voltage-gated calcium entry would be reduced, and calcium removal from the cell facilitated via sodium/calcium exchangers and/or P-type calcium ATPase pumps located in the plasmalemma [71]. What had intrigued me was a report that hypoxia attenuated this response in pulmonary arteries, as it seemed plausible that inhibition of this pathway to vasodilation might facilitate HPV by removing opposition to depolarisation-evoked calcium influx consequent to the inhibition of IK_N_ and IK_V_. Consistent with previous studies, we found that β-adrenoceptor activation and membrane permeable cAMP analogues did indeed hyperpolarise pulmonary arterial myocytes by activating iberiotoxin-sensitive BK_Ca_ channels, and in a PKA-dependent manner. Critically, vasodilation by β-adrenoceptor activation was attenuated by ~60% by blocking RyRs with ryanodine, ~60% following depletion of SR calcium stores through SERCA inhibition using cyclopiazonic acid (Figure 3), and by ~90% following direct block of BK_Ca_ channels with iberiotoxin [25]. In short, vasodilation was indeed facilitated by SR calcium release through RyRs and consequent activation of BK_Ca_ channels.

## 5. Hey Hey CPA Say: Had Me a Store but One Ran Away

We had confirmed that two opposing responses, vasodilation and vasoconstriction, were mediated by calcium release from the SR through RyRs. Moreover cAMP-dependent vasodilation was facilitated by calcium release from a cyclopiazonic acid-sensitive SR store. There is only one contiguous unit of SR [13,72] and cyclopiazonic acid is a non-selective SERCA inhibitor [73,74], right? So, HPV would be even more sensitive to pre-incubation with cyclopiazonic acid, particularly when the endothelium is removed from the arteries. Stands to sense, under these conditions we had shown HPV to be entirely dependent on SR calcium release through RyRs. Right question, wrong answer, again!

Cyclopiazonic acid abolished the first transient phase of HPV, but left the sustained phase 2 constriction unaltered, the second phase reaching a steady plateau much like control, that was sustained for hours [24]. My response to Michelle was, “it doesn’t make sense, do it again”. Same thing happened, again and again (Figure 4). Adding to this, in case it has escaped your attention, our studies on vasodilation had relied upon pre-constriction of pulmonary arteries with prostaglandin F-2α, which we had previously confirmed to induce vasoconstriction, at least in part, by mobilising intracellular calcium stores [41].

Could there be two discrete SR stores after all, one sensitive and one insensitive to cyclopiazonic acid? There were data in the literature to support such a position [75,76,77,78,79], and the earliest of these studies was on pulmonary arteries; albeit from the guinea-pig, which exhibits markedly attenuated HPV due to adaptation to life at altitude [80]. Moreover, much like previous studies on acutely isolated airway smooth muscles [81,82], we had found that in acutely isolated pulmonary arterial myocytes isoprenaline-induced signalling pathways preferentially raised intracellular calcium concentration at the perimeter of these cells, proximal to the plasmalemma [25].

## 6. 8-Bromo-cADPR and the Magical Mystery Tour Takes Off

During 1997 I became aware of a certain β-NAD^+^ metabolite, cADPR, from a poster I read while wandering the corridors at Oxford. Between 1989 and 1991, Hong-Cheung Lee and co-workers had published their seminal studies on sea urchin eggs that led to the discovery of cADPR, and the enzyme activities for its synthesis and metabolism [21,83,84]. More importantly still, from my perspective at the time, they had also identified the most likely downstream target of cADPR, namely sea urchin RyRs [85]. Albeit very slowly, data had thereafter begun to emerge in support of a similar role for cADPR in mammalian cells [84,86,87,88,89,90]. A competitive and membrane permeable cADPR antagonist had also been developed and was made available to me, namely 8-bromo-cADPR (Figure 1) [20]. Given its likely role in regulating RyRs, it seemed more than plausible that cADPR might facilitate HPV or pulmonary artery dilation, but it couldn’t be involved in both now, could it?

To break the suspense, 8-bromo-cADPR did indeed block isoprenaline-induced dilation of pulmonary arteries (Figure 3) [25], and also inhibited HPV in isolated pulmonary artery rings and in the rat lung in-situ (Figure 4) [24]. However, with respect to the latter its effects were quite different from the picture painted by the combinatorial action of ryanodine and caffeine, which blocked HPV in its entirety [41].

In arteries with and without endothelium, pre-incubation with 8-bromo-cADPR had no effect on phase 1 of HPV, which was blocked by pre-incubation with ryanodine and caffeine. But, somewhat counterintuitively, it abolished the sustained phase 2 constriction whether the endothelium was present or not (Figure 4) [24]. Once again, I told Michelle that the experimental outcomes did not make any sense, and that she would have to repeat this experiment again, but this time in a variety of different ways. We’ll come back to that later, suffice to say for now that 8-bromo-cADPR blocked phase 2, but not phase 1 of HPV. This was in stark contrast to cyclopiazonic acid, which had taken out the phase 1 constriction but left the sustained phase 2 constriction unaltered (Figure 4) [24]. In short, with respect to HPV the effects of inhibiting RyR activation with 8-bromo-cADPR and blocking SR calcium uptake with cyclopiazonic acid were not simply opposite from each other, but contrary. These findings appeared incompatible with the view that the SR operated as one contiguous unit [13]. Yet the effects of pre-incubation with ryanodine and caffeine were consistent with a contiguous SR, given that HPV and constriction by prostaglandin F-2α were blocked in their entirety under these conditions [41]. Recall again, that for studies on cADPR-dependent vasodilation we pre-constricted pulmonary arteries with prostaglandin F-2α, which induced vasoconstriction, in part, by mobilising SR calcium from a cyclopiazonic-insensitive store [41].

Accepting all of the above to be bona fide, it was evident that if the effects of 8-bromo-cADPR on HPV were indeed due to it blocking the action of cADPR, then cADPR must be necessary for SR calcium release to occur during the initiation and maintenance of phase 2 of HPV, even though it was not required in the context of phase 1. For this to be the case hypoxia must somehow increase cADPR levels in pulmonary arterial smooth muscles. This was confirmed by parallel analysis of the synthesis and metabolism of cADPR within homogenates of pulmonary arterial smooth muscles, with the support of the Galione laboratory through the redeployment of Heather Wilson and Justyn Thomas; a favour to be repaid, and some, a few years later. Surprisingly, these endogenous enzyme activities were greater in smooth muscles from pulmonary arteries when compared to that from systemic arteries. This was a significant finding because it provided, in some small part, the necessary degree of pulmonary selectivity that would be required of any mediator of HPV. Moreover, the level of ADP-ribosyl cyclase and hydrolase activities were inversely related to pulmonary artery diameter [91]. Accordingly, analysis of smooth muscle cADPR levels revealed marked increases in its accumulation during hypoxia (16–21 mm Hg; 1 h). Twofold in second-order branches of the pulmonary arterial tree, and tenfold in third-order branches. Thus, much like the magnitude of HPV and the distribution of the enzyme activities for cADPR synthesis, hypoxia-evoked increases in cADPR content were inversely related to pulmonary artery diameter [91].

Our further studies provided evidence that increased β-NADH formation under hypoxic conditions may facilitate cADPR formation from β-NAD^+^, by either augmenting ADP-ribosyl cyclase and/or inhibiting cADPR hydrolase activities [91]. Adding to this, later experiments suggested that cADPR accumulation and/or RyR activation by cADPR requires prior activation of AMPK, consequent to inhibition of mitochondrial oxidative phosphorylation during hypoxia [49,50]. Over and above this, the precise mechanism by which hypoxia promotes cADPR accumulation in pulmonary arterial smooth muscles remains to be confirmed.

## 7. 8-Bromo-cADPR Action: On the Face of It All or Nothing

The aforementioned studies led us to perhaps the most unexpected and surprising observation thus far. As mentioned above, when presented with the records showing that 8-bromo-cADPR blocked phase 2, but not phase 1, of HPV, I told Michelle that these experimental outcomes did not make any sense, and suggested she repeat this experiment again, and in a variety of different ways. This was Pharmacology we were dealing with, not Physiology, and 8-bromo-cADPR was purportedly a competitive antagonist. Therefore, to understand its mechanism of action on pulmonary arteries in necessary detail, a concentration-response curve would be required, with and without the endothelium. My general viewpoint was that the differential effects on phase 1 and phase 2 of HPV must in some way be related to the concentration applied. Either we were dealing with differences in affinity between different RyR subtypes, or the pharmacokinetics of 8-bromo-cADPR led to difficulties in accessing one cellular compartment relative to another.

As ever Michelle received the plan with a smile, and promptly got on with it. We reconvened to discuss her findings sooner than I had expected, within one week. She presented me with yet another imponderable. When arteries were pre-incubated with 8-bromo-cADPR at 1μM there was no effect on either phase 1 or phase 2 of HPV, but at 3 μM phase 2 was abolished (Figure 5). All-or-none block with a competitive antagonist? Quite mad! I began to doubt the compounds legitimacy, as not even phenoxybenzamine would act like this over the time period given for pre-incubation; for the uninitiated, phenoxybenzamine binds to its protein targets (e.g., alpha adrenoceptors) covalently, irreversibly and in a time-dependent manner. A competitive antagonist should block a response driven by agonist-receptor coupling in a concentration-dependent manner, with complete block achieved 1–2 orders of magnitude above the threshold concentration at which partial inhibition of this response is first seen.

Could we reveal concentration-dependency of 8-bromo-cADPR in another way? I recalled the all-or-none, quantal activation of units of nicotinic acetylcholine receptors by acetylcholine at the neuromuscular junction, and the demonstration that competitive antagonists of nicotinic acetylcholine receptors could block neuromuscular transmission in an all-or-none manner when applied prior to induction of a single “spike” of contraction, yet deliver concentration-dependent inhibition of contraction once tetanus (sustained contraction) had been induced [92,93,94,95,96]. The all-or-none block of neuromuscular transmission occurred when the activation of a “critical mass” (~45%) of nicotonic acetylcholine receptors could no longer be achieved. With this in mind I decided that we should test the effect of 8-bromo-cADPR after prior induction of HPV in arteries without endothelium (to remove confounding effects of endothelium derived vasoactive agents). Once initiated, the maintained phase 2 constriction was, to my relief and excitation, reversed by 8-bromo-cADPR in a concentration-dependent manner. The threshold concentration for partial inhibition of HPV was 3 μM, which delivered all-or-none block by pre-incubation, while complete reversal of HPV was achieved at 100 μM (Figure 5). As indicated above, outcomes were analogous to the action of competitive antagonists at the skeletal neuromuscular junction, that told us so much about the importance of junctional complexes to intercellular signalling [95,97,98,99], and where neuromuscular transmission is compromised once a critical mass of skeletal muscle nicotinic acetylcholine receptors are blocked. Could there be an intracellular junction of the SR that is critical to the induction of HPV, in which RyRs play a role similar to that of junctional nicotinic acetylcholine receptors? We concluded that our data for 8-bromo-cADPR were consistent with this idea. We proposed that a similar “margin of safety” may therefore be built into HPV through signal transmission across some form of intracellular junction, and in a manner coordinated through the activation of a subpopulation of RyRs. We suggested [24] that this might allow for the cADPR-dependent component of HPV to be initiated in an all-or-none manner, and thus offer a plausible explanation for the all-or-none block of HPV by 8-bromo-cADPR, providing its primary action was to block:

“the activation by cADPR of a certain proportion of RyRs” that is critical to the initiation of HPV. and

“cADPR-dependent calcium mobilisation from a sub-population of RyRs” that serve a unique and specific function in the context of HPV.

This was pertinent to the wider picture derived from further consideration of isoprenaline-induced vasodilation and HPV, which led to the following postulates:

“β-adrenoceptor signalling targets cADPR synthesis to a particular (read distinct) RyR subtype in the “peripheral” SR that is in close apposition to BK_Ca_ channels in the plasma membrane” [25];

“cADPR-dependent vasoconstriction results from the activation of a discrete RyR subtype localised in the “central” SR…” [25];

An SR compartment... “in close apposition to the plasma membrane, (is) served by a SERCA pump that is sensitive to cyclopiazonic acid” [23];

An SR compartment...“in close apposition to the contractile apparatus is served by a SERCA pump that is insensitive to cyclopiazonic acid” [23];

Thereafter I proposed that [23]:

Phase 1 of HPV might be mediated by the mobilisation of calcium from an SR compartment served by a cyclopiazonic acid-sensitive SERCA, that is inhibited by hypoxia due to a fall in ATP supply proximal to the plasma membrane.

The phase 2 constriction driven by cADPR-dependent SR calcium release therefore requires the presence of a second, spatially segregated SR calcium store that is served by a discrete, cyclopiazonic acid-insensitive SERCA pump that recycles calcium locked into the cell by inhibition during hypoxia of the cyclopiazonic acid-sensitive SERCA operating at SR juxtaposed to the plasma membrane.

Only time and further investigations would determine whether or not this was indeed the case.

## 8. Location, Location: Dispense with the Wax and Shine Some Light on Here

In certain cell types, such as smooth muscles, evidence had long suggested that at least two spatially segregated and independently releasable subcompartments of calcium may exist within the SR network, perhaps supplied by different, pharmacologically distinct types of SERCA pump [75,76,77,78,79]. Moreover, previous studies on acutely isolated airway smooth muscles [81,82] and our own similar studies on pulmonary arterial myocytes [25] suggested that β-adrenoceptor activation preferentially raised intracellular calcium at the perimeter of cells, proximal to the plasmalemma. However, such proposals had gained little traction, even though emerging evidence indicated that a single cell type could express SERCA2a, SERCA2b [100] and a wider variety of intracellular calcium release channels than previously thought. In some cases all three RyR subtypes were present [101,102], in combination with one or more of the three IP_3_R subtypes [25,77,103].

I considered this information in the context of the pharmacological insight provided by 8-bromo-cADPR, and decided to explore the possibility that the functional segregation of SR calcium stores could be very effectively achieved through the deployment of spatially segregated subtypes of RyR and SERCA. At this point I was supported by a new crew at the bench, namely Nick Kinnear and Jill Clark, and further aided by the kind gift during 2002 of affinity purified sequence-specific antibodies against RyR1, RyR2 and RyR3 from Sidney Fleischer, and sequence-specific antibodies against SERCA2a and SERCA2b from Frank Wuytack. With these tools at our disposal we then developed semi-quantitative approaches by which we could assign spatial locations to each RyR and SERCA subtype. Briefly, we analysed the relative density of labelling for each within one of three designated regions of the cell, the subplasmalemmal region (within 1μm of the plasma membrane), the perinuclear region (within 1μm of the nucleus) and the extra-perinuclear region (everything that lay in between).

### 8.1. The SERCA Circus

One could not have wished for more strikingly different outcomes for SERCA2a and SERCA2b. Our analytical nous was hardly required to draw a conclusion. Visual inspection of deconvolved Z sections and three-dimensional (3D) reconstructions of immunofluorescence labelling was sufficient. SERCA2a was restricted to perinuclear regions, while SERCA2b was located proximal to the plasmalemma (Figure 6).

Semi-quantitative analysis of their density of labelling by region simply confirmed what we could see with our own eyes [103]. The vast majority of SERCA2b labelling (~70%) lay within the sub-plasmalemmal region, with ~8% extra-perinuclear and ~20% perinuclear. In marked contrast, SERCA2a labelling was almost entirely (~90%) restricted to the perinuclear region. Therefore, native SERCA2b must feed the SR proximal to the plasma membrane, where cADPR must facilitate the calcium-dependent component of vasodilation in response to β-adrenoceptor activation (Figure 7). If so, then SERCA2b must represent the cyclopiazonic acid sensitive SERCA, perhaps due to its location proximal to the plasmalemma rather than by any selective pharmacological action *per se*. By contrast, SERCA2a clearly supplies deeper, perinculear SR and most likely represents the cyclopiazonic acid-insensitive SERCA that supplies the component of SR critical to HPV; perhaps spared simply because the duration of pre-incubation allowed was insufficient to attain an effective concentration anywhere except at the periphery of the cell [103].

So, why might hypoxia mobilise calcium from two discrete SR compartments? Well, it appeared that the same cyclopiazonic acid-sensitive SR store was utilised during phase 1 of HPV and vasodilation in response to β-adrenoceptor activation. It is possible, therefore, that SR calcium release by hypoxia serves two purposes. Hypoxia may primarily trigger constriction by calcium release from a central SR compartment that is in close apposition to the contractile apparatus and served by a cyclopiazonic acid-insensitive SERCA pump (SERCA2a). A secondary action of hypoxia could be to deplete and/or block filling of a peripheral SR compartment by inhibition of a cyclopiazonic-acid-sensitive SERCA pump (SERCA2b) that lies in close apposition to the plasma membrane, and normally facilitates vasodilation by removing cytoplasmic calcium to the peripheral SR (Figure 7). This could explain why pulmonary artery dilation by β-adrenoceptor activation is attenuated by hypoxia [104], and HPV is enhanced by cyclopiazonic acid [105] yet abolished by SERCA inhibition with thapsigargin [106].

### 8.2. To Be Three RyRs

Analysis of the distribution of RyRs by subtype was a little more problematic, as they were not so cleanly separated. Adding to this, two years into our study I discovered to my surprise, that Sid Fleischer had provided James Sham with samples of his RyR antibodies, for the same purpose as that which I had outlined to him some years earlier. Sid apologised for this oversight, caused by his forgetting the precise nature of our project; which perhaps argues in favour of completing material transfer agreements even with supporters and friends. Thankfully, James’ laboratory completed a qualitative analysis of labelling in primary cultures of pulmonary arterial myocytes [107], which did little more than confirm earlier studies that had identified the expression of all three RyR subtypes in arterial myocytes [101,102]. As we will see below, cultured myocytes are an altogether different model when compared to acutely isolated cells. Therefore, Nick and Jill, independent of each other, ploughed on with their exhaustive, comparative analyses of RyR subtype distribution which confirmed the following outcomes [103,108].

By density of labelling (Figure 8) RyR1 was the primary subtype targeted to the sub-plasmalemmal region of pulmonary arterial myocytes (3–5-fold higher levels than RyR2 and RyR3). This is consistent with a role for RyR1 in facilitating vasodilation in response to β-adrenoceptor activation. Accordingly, our findings and those of others suggested that of the available RyRs, RyR1 is most sensitive to activation by cADPR [86,109,110]. By contrast, cADPR does not appear to activate RyR2, rather it increases the sensitivity of RyR2 to activation by calcium, and thus facilitates signal propagation along arrays of RyR2 clusters by calcium-induced calcium release (CICR) [86,90,110]. RyR3 is, much like RyR1, activated by cADPR, albeit at slightly higher concentrations [109]. However, we detected little or no RyR3 within 1μm of the plasmalemma, which argues strongly against a role for RyR3 in facilitating calcium release from subplasmalemmal SR in support of vasodilation. We gained further insights from the studies of others on the kinetics of RyR activation. Under matched conditions the threshold for activation of RyR1, RyR2 and RyR3 appears similar, with channel activation when calcium concentrations at the cytoplasmic surface exceed 100nM [111,112,113]. However, RyR1 is relatively insensitive to CICR. It exhibits relatively little gain in probability of opening (0–0.2) with increasing cytoplasmic calcium concentration, while mean open times increase only twofold over the activation range for RyR1 [111,112,113]. Adding to this the IC_50_ for inactivation of RyR1 is ~1 μM [111,112]. Moreover, it is evident that propagation of calcium signals away from discrete RyR1 clusters can be limited by proximal RyR1 binding partners, the organisation of the SR and proximal plasmalemma, and the presence or absence of structures that span these two membranes (for review see [113,114]). Therefore, all things considered, it appeared most likely that cADPR-dependent activation of RyR1 offered the most likely path through which SR calcium flux might recruit plasmalemmal BK_Ca_ channels, to thus deliver membrane hyperpolarisation, calcium removal from the cell (SR emptying) and ultimately vasodilation. That said, significant levels of RyR1 were present in the extra-perinuclear region of pulmonary arterial myocytes, and to a lesser extent in the perinuclear region. It was therefore evident that RyR1 may also contribute in some way to the regulation of calcium signalling in other regions of the cell, a matter which we will return to later.

RyR2 was found to be primarily located within the wider extra-perinuclear region of pulmonary arterial myocytes (Figure 8). The density of labelling for RyR2 within this region was 3–4 fold higher than that for RyR1 and RyR3, and fell 2 and 5 fold either side in the subplasmalemmal and perinuclear regions, respectively [103,108]. This appeared consistent with a role for RyR2 in signal propagation and myocyte contraction, particularly when one considers excitation-contraction coupling in cardiac muscles, which is so precisely coordinated by RyR2 clusters that carry propagating calcium waves by CICR [5,9,113,115]. Critical in this respect is the 0.5μm spacing between RyR2 clusters, its relatively low EC_50_ for CICR, low sensitivity to inactivation by calcium and high gain in open probability with increasing calcium concentration, which together ensure that once initiated a propagating calcium wave is less prone to failure than when carried by other RyR subtypes [111,112,113,115]. We concluded, therefore, that RyR2 most likely functioned to carry propagating global calcium waves in support of contraction in pulmonary arterial smooth muscles. Supporting this, as mentioned above, it is evident that cADPR does not activate RyR2 *per se*, rather it facilitates CICR via RyR2 by increasing its sensitivity to activation by calcium, which may facilitate the propagation of calcium waves once initiated [86,110].

This left us with a clear question to resolve. In the context of HPV and in the absence of calcium influx from the extracellular milieu, where could the trigger calcium necessary for stable RyR2 recruitment arise from, and with the required margin of safety?

In this respect, I excluded RyR1 as the source of trigger calcium for little good reason, other than our conclusion that subplasmalemmal RyR1s most likely facilitated calcium-dependent vasodilation. This left RyR3. Significantly, the density of RyR3 labelling declined markedly (between 4- and 14-fold by region) outside the perinuclear region of the cell (Figure 8 [103,108]). Irrespective of the mechanism of RyR3 activation, therefore, once recruited it seemed unlikely that RyR3 would function to carry a propagating calcium wave far beyond the point of initiation. This sat well with the marked increase in density of labelling for RyR2 within the extra-perinuclear region when compared to the perinuclear region. In short, RyR2 might function to receive calcium from RyR3. A plausible solution, given that a discrete signalling pathway for activation of perinuclear RyR3s could be delivered by regions of the plasmalemma that are necessarily juxtaposed to perinuclear SR where the nucleus sits. But how might RyR3 be differentially activated?

In 2004 we were presented with one possibility, through the discovery by Nick Kinnear and I of lysosome-SR nanojunctions in pulmonary arterial myocytes (Figure 9 [116,117]), during investigations that were informed by the previous publications of Grant Churchill on lysosome-ER coupling in sea urchin eggs [118,119]. Lysosome-SR nanojunctions appeared to be critical to the action of the second calcium mobilising pyridine nucleotide previously discovered by HC Lee and co-workers, nicotinic acid adenine dinucleotide phosphate (NAADP) [120], and were primarily formed by the interaction of dense lysosome clusters with perinuclear regions of the SR that were rich in RyR3 (Figure 9 [108]).

In short, it appeared that lysosome-SR nanojunctions provided a trigger zone, or intracellular synapse, for induction of calcium signals. Accordingly, we established that SR resident junctional RyR3s are activated when lysosomal calcium stores are mobilised by NAADP, with extraperinuclear RyR2s likely required to carry propagating calcium waves beyond this perinuclear region [108,116].

While presenting these findings to the great and the good of Robert Wood Johnson Medical School in New Jersey during 2005, my host and good friend Jianjie Ma suggested that he might have a good collaboration for me. On returning to the UK a conference call was organised by Jianjie, during which I was introduced to Mike Zhu, who had cloned Two Pore Channel 2 (TPC2) some years earlier and demonstrated that it was targeted to lysosomes. Mike had then hit a brick wall, due to the fact that he had no way of examining the functional role of TPC2 inside cells. My lab offered the necessary approaches, namely intracellular dialysis of second messengers from a patch pipette and parallel calcium imaging. Mike promptly supplied HEK293 cells that stably over-expressed human TPC2. With this model at my disposal and two new recruits to deploy, namely in Peter Calcraft (PhD student) and Chris Wyatt (post-doc), my lab became the first to demonstrate that NAADP gated lysosomal calcium release in a TPC2-dependent manner, which was confirmed using siRNA knockdown of TPC2 and supporting pharmacological interventions [121,122]. I then, with Mike’s prior approval, invited Antony Galione to join our collaborative team, with a view to his lab confirming that TPC2 was an NAADP receptor by radioligand-binding assay. I was aware that Antony had been looking for a candidate NAADP receptor for years, so it seemed only right to repay him for his previous support of my work. The rest, one could say, is another history [123,124,125].

Could NAADP-dependent activation of TPC2 within lysosome-SR nanojunctions offer a logical solution to our conundrum? Well yes, providing hypoxia triggers lysosomal calcium release through TPC2, which is then amplified through RyR3 activation by CICR, such that a propagating calcium wave is initiated by all-or-none recruitment of arrays of RyR2 clusters that span the wider cell, and in a manner that could be facilitated by cADPR. Indirect support for this view was provided by the identification of frequent discharge sites (FDS) in the perinuclear region of visceral and vascular myocytes, which were generally devoid of mitochondria [126], but, as we had now shown, replete with large lysosome clusters [108,116]. A role for lysosome-SR nanojunctions seemed all the more plausible when mTORC1, which is inhibited by AMPK, was implicated in the progression of hypoxic pulmonary hypertension [127,128]. Unfortunately, however, science is never that simple. Our subsequent unpublished studies indicated that the putative, non-selective NAADP antagonist NED-19 did not affect hypoxia-evoked constriction of isolated pulmonary artery rings, while others found that HPV remained unaltered after pre-incubation with two putative, non-selective NAADP antagonists [59]. Adding to this, using spectral Doppler ultrasound we recently found acute HPV in-vivo in mice to remain unaffected following global deletion of the gene (*Tpcn2*) that encodes TPC2 (unpublished). In short, if TPC2 plays a role in acute HPV it is a subtle one. Alternatively, there may be redundancy of function within the system, perhaps through the expression of alternative lysosome-resident, calcium permeable channels such as TRPML1 [129,130,131,132] and TRPM2 [133,134], each of which has been proposed to couple to RyRs and impact smooth muscle function and vascular reactivity. In this respect it is notable that mTORC1-dependent modulation of autophagy impacts pulmonary vascular remodeling [127,128], given that inhibition of mTORC1 [135] may regulate lysosomal TPC2 [121] and TRPML1 [136]. Therefore, while lysosomal calcium flux may not be required for HPV, it likely contributes to the development of hypoxic pulmonary hypertension.

If neither TPC2 or lysosomal calcium flux is required for induction of acute HPV, could activation of RyR3 by cADPR be key to the all-or-none block of HPV by 8-bromo-cADPR? Well, we now know that RyR3 is activated by cADPR, and at slightly higher concentrations than required to activate RyR1 [109]. Accordingly, intracellular dialysis from a patch pipette of low concentrations of cADPR increases cytoplasmic calcium concentration at the perimeter of pulmonary arterial smooth muscles [25], while higher concentrations of cADPR initiate a global calcium transient [135]. Therefore, an alternative explanation could be that cADPR-dependent activation of junctional clusters of RyR3 might be critical to the induction of HPV, through subsequent recruitment by CICR of extra-junctional arrays of RyR2 clusters. If RyR3 clusters are organised into discrete active units or couplons (for review see [113]), then this could indeed explain the all-or-none block of HPV by 8-bromo-cADPR. Once HPV is initiated, cADPR accumulation could then act to facilitate regenerative, propagating calcium waves by sensitising RyR2s to CICR. Blocking the latter could plausibly explain the concentration-dependent reversal of HPV by 8-bromo-cADPR, following its initiation. In short, activation of perinuclear RyR3 couplons alone could provide a “margin of safety” for all-or-none recruitment of extra-junctional RyR2 clusters and reduce the probability of false events being initiated. Moreover, such a model would allow discrete, cADPR-dependent activation of peripheral RyR1s in support of vasodilation.

While the studies of others have confirmed the regional distribution of SERCA and RyRs in pulmonary arterial smooth muscles [137] (see also airway smooth muscles [138] and cerebral artery smooth muscles [139]), and the critical role in HPV of SR calcium release via RyRs [59,105,140], outcomes for studies using knockout mice have been perplexing. Briefly, gene deletion strategies have suggested a prominent role in HPV for RyR1 [141], RyR2 [142] and RyR3 [143]. A role in HPV for all three RyR subtypes seems highly unlikely, but one never can tell, especially when so many studies fail to distinguish between phase 1 and phase 2 of this response. So, we have arrived at one plausible mechanism, but without critical evidence in support of a role for lysosome-SR nanojunctions, the nature of the junctional complex involved remains elusive.

## 9. Another Twist in the Tale

In support of our immunofluorescence labelling we carried out RT-PCR and Western blots, supported by the forever positive Gordon Cramb, and his lucky white heather. RT-PCR identified transcripts for SERCA1, SERCA2a, SERCA2b and SERCA3 in whole arteries without endothelium [103]. By contrast, our sequence-specific antibodies identified only SERCA2a and SERCA2b on Western blots of pulmonary artery homogenates that excluded the nuclear fraction, in agreement with previous studies on other vascular smooth muscles [100]. Jill Clark had, however, proceeded with immunofluorescence assays for all SERCA using acutely isolated pulmonary arterial myocytes. During 2004 she presented me with a number of beautiful images of 3D reconstructions of DAPI labelled nuclei that identified odd bands of labelling for SERCA1 across the outer surface of the nucleus (three animals, n = 10 cells). No labelling was evident in any other part of the cell, so I dismissed this as non-specific labelling, much to Jill’s evident disappointment. A couple of months later I was to eat my words.

During one particular lab meeting Nick Kinnear was taking me through his data analysis on RyR1 distribution in 3D reconstructions of pulmonary arterial myocytes. He had applied a digital skin/mask to better highlight RyR1 distribution and removed all labeling smaller than 100 nm^3^. The bar chart (mean ± SEM) in Nick’s Excel spreadsheet indicated a massive amount of RyR1 in the perinuclear region of the cell, yet there appeared to be little or no labelling in the perinuclear region of the 3D image of the example cell he had chosen. I pointed this out to him, but he struggled to dismiss the data in the spreadsheet. I interrupted Jill’s daydream by asking her to have a look, “Do you understand where my concern lies?”. She said yes, and I told her to explain it to Nick, find out what the discrepancy was and then come and find me. Sometime later, maybe a week but no more, Jill appeared in my office and told me they had something to show me. I told her I was about to give a lecture, my adrenaline was up, and that it was perhaps not the best time. She insisted, and said I would be happy with what they’d found. She was right but I was also perplexed, yet again.

Nick and Jill had created a movie in which the cell labelled for RyR1 was rotated, the DAPI channel removed, and then the cell rotated once more (Figure 10). This revealed a tubular network of labelling within the boundary of the nucleus. I recalled the SERCA1 labelling Jill had shown me earlier. They were ahead of me, and the movie revealed a similar tubular network of labelling within the nucleus (Figure 10). No such network was observed in any cell labelled for RyR2, RyR3, SERCA2a or SERCA2b. I asked Jill if she’d retained the nuclear fraction removed from the homogenates used for her previous Western blots. Like all good scientists she had, and when this fraction was run it revealed a very faint band for SERCA1 [144].

Jill went on to load the lumen of the SR with Calcium Orange, which revealed a clear tubular network that appeared to span the nucleus within living cells, but there was no membrane permeable marker for the nucleus of living cells available at this time. I didn’t know how to take this work forward, so I decided to exclude any comment on nuclear labelling from our subsequent publications [103,108], “sat back” and waited for a day when I could further explore this signalling domain, hoping that no-one would beat me to the punch.

By chance, that same year I was invited to give a talk at a symposium in honour of the pending retirement of Tom Bolton and Alison Brading, both of whom had been very kind in their advice and guidance to me, during my time at the Pharmacology Departments of St George’s Hospital Medical School and Oxford, respectively. I was thrilled to be invited and sought to put together my best slide presentation yet. It was a fortuitous moment in time, because during one coffee break I was approached by an elder Dutch gentleman, Casey van Breemen. Casey said he liked my talk, suggested that we collaborate, and asked me to drop him an e-mail after the conference had finished. Due to my circuitous route into the field of smooth muscle calcium signalling (Oxford, and beyond for that matter) I had no detailed knowledge of Casey’’s work, even though I had sat through his talk, all too blindly focused on the delivery of my own. However, Casey had given me his card, and I later e-mailed him. I then realised he was based in Vancouver, where I had booked a holiday ahead of Experimental Biology 2006. I visited UBC Vancouver, gave a seminar, and met Nicola Fameli for the first time. Then, in 2008 Casey invited me out to Vancouver again, and this time paid for my flight and accommodation. During this visit I decided to show Casey and Nick the nuclear labelling for RyR1 and SERCA1. Casey advised me not to show this to anybody else before a paper was submitted, not that I needed any such advice.

## 10. We Will Be Fine, Its Nano Time: Even though “NAScA” Say We Out of Line

I had learned that Casey, Nick Fameli and I shared a common interest in segregated signalling domains within cells, Casey having been the first to characterise plasma membrane (PM)-SR nanojunctions in smooth muscles (for historical perspective see [145,146], while I had conceived of and identified lysosome-SR nanojunctions while pursuing our independent evidence of calcium signalling within discrete cytoplasmic “microdomains” (for historical perspective see [23]). Nick Fameli was not a biologist, but a physicist with expertise in computer modelling who had been employed by Casey to apply such modelling in the further examination of how PM-SR junctions may function to adjust the calcium load within the SR lumen, by studying “what we could not yet see”. We further developed our joined-up thinking during (for me) late night Skype sessions from my office, now at the University of Edinburgh, while Casey and Nick enjoyed their morning coffee. During these discussions I was introduced to considerations of not microdomains, but cytoplasmic nanodomains for calcium signalling. On one occasion Casey could not make the allotted Skype time, so Nick Fameli took me through his latest modelling of calcium exchange across cytoplasmic nanodomains. He was explaining that once a junctional membrane pair was separated by more than 50 nm junctional integrity weakened in a manner that could be compensated for by increasing the number of SERCA pumps, but above 200 nm the capacity for site-specific calcium exchange was not just reduced but lost entirely. I cut him short to ask “Where does the calcium go then?”. Nick replied “it diffuses out of the sides”. To be precise, the probability of random diffusion of calcium away from the junction formed by the membrane pair was higher than that for capture by SERCA when the junctional membrane separation was increased from 200 nm to 400 nm, irrespective of the density of SERCA pumps in the SR membrane [147,148]. Eureka, it was not just the “nanodomain” but the “nanojunction” that was critical to our thinking! Nanojunctions not only acted to direct calcium signalling, but also restricted the random diffusion of calcium away from the cytoplasmic space demarcated by them. We hastily arranged another Skype meeting with Casey to discuss this “revelation”. Our ongoing discussions led to the “Panjunctional Sarcoplasmic Reticulum” hypothesis that was framed in an invited review for the Journal of Physiology [22], which, at my insistence, excluded any consideration of nuclear calcium signalling. In essence this hypothesis was built around the proposal that cellular membrane-membrane nanojunctions are formed by the SR at defined “target sites” to deliver highly localised and functionally segregated calcium signals, with the functional specification of a given signal determined both by the constraints on diffusion imposed by the nanojunction itself and by the calcium signalling machinery incorporated within a given junctional complex. Nanojunctions are defined by their distances of separation and had been identified not only between the S/ER and the PM, but a variety of organelles including lysosomes, mitochondria and the nucleus [22]. We knew at the time that the underlying mechanisms of signal generation were likely more elaborate in nature than we could possibly envisage, but evidently relied on the strategic targeting to their designate nanojunction of macromolecular complexes that incorporate different types of calcium transporters and release channels, each of which may be characterised by different kinetics and affinities for calcium, and may in turn be differentially modulated by their respective binding partners, second messengers and enzymes [103]. This is self-evident when one considers, for example, the fact that in arterial and arteriolar smooth muscles alone, we now know that intracellular calcium signalling is coordinated by the gating of:

Two of the three known S/ER resident IP_3_ receptors (IP_3_R1-3) by inositol 1,4,5 trisphosphate (IP_3_) [149,150,151,152,153];

Up to three S/ER resident ryanodine receptor subtypes (RyR1-3) [101,107,108,153] by calcium and/or cADPR [24,25,91,108,109,117,154];

Endolysosome targeted two pore channel (TPC) subtypes 1 and 2 by either NAADP [116,117,122,135,155], phosphatidylinositol-3,5-bisphosphate (PI3,5P_2_) [156], or mTORC1 [135,157];

Endolysosome targeted TRPML1 [129,130,131,132] and TRPM2 [133,134] by mTORC1 and/or ADP-ribose;

The mitochondrial calcium uniporter [158,159].

Furthermore, the expression pattern of calcium release channels and pumps in smooth muscles may vary between regions in the same vascular bed [153] and from one vascular bed to another [103,138,153,160,161,162]. In short, variations in the prevalence of nanojunctions, their ultra-structure and their respective molecular machinery could explain, in part, both functional heterogeneity [25,153,160,161,163] and plasticity [22,137] of smooth muscles, and wider differences evident between smooth muscles and other cell types [13].

With regard to the likely importance of nanojunctions themselves to cellular communication, we need look no further than the neuromuscular junction, where the critical role of nanojunctions in intercellular communication has been evident for nearly a century [11]. It is the organisation of these junctional membranes that proved to be so critical to neuromuscular transmission through their coordination of the release, receptor interactions and reuptake of acetylcholine [95,97,98,99]. Here, electron micrographs indicate that the pre- and post-junctional membranes are approximately 20 nm apart, and extend more or less parallel to each other over several hundred nm. It is important to note, however, that junctional dimensions seen within electron micrographs may be smaller than they are in reality, due to sample dehydration.

Given the above it is surprising that so little attention has been paid to the role in intracellular signalling of the plethora of junctions formed between intracellular membranes, beyond acknowledging that there are “contact sites” that may aid the direction of ion fluxes, exchange or the action of other messengers. The one notable exception is of course in the process of excitation-contraction coupling in skeletal and cardiac muscles, where the junctional complexes formed between T-tubules of the sarcolemma and terminal cisternae of the SR are well documented, and are equally critical to the coordination of excitation-contraction coupling as the neuromuscular junction itself. In each instance electron micrograpahs suggest that the junctional membrane pair are separated by ~20 nm [164,165,166,167,168]. There is, however, one important distinction between skeletal and cardiac muscles that must be mentioned in terms of the mechanism by which calcium is mobilised during excitation-contraction coupling, and this will undoubtedly have resonance with respect to our future understanding of the versatility of signalling across all intracellular nanojunctions. Against earlier predictions, it is now evident that the sarcolemma-SR nanojunctions of skeletal muscle allow for the transfer, through non-covalent association, of electrostatic charge between sarcolemma resident dihydropyridine receptors and SR resident RyR1s (see for example [169,170,171]), which ultimately gates calcium release from the SR via RyR1 couplons [113]. By contrast, the sarcolemma-SR nanojunctions of cardiac muscle support what is generally regarded as the classical form of junctional coupling, namely agonist-receptor interactions, by targeting calcium influx to RyR2 clusters located on the terminal cisternae of the SR which, in turn, trigger a propagating calcium wave and contraction by further CICR from the SR via arrays of RyR2 clusters, each RyR2 cluster separated from the other by ~500 nm [9,115]. Whatever the mechanism of transduction, electrostatic or agonist-receptor gating mechanisms, these sarcolemma-SR junctions represent the archetypal intracellular nanojunctions, with each one being separated by ~20 nm (not allowing for reductions through dehydration) and clearly designed to accurately deliver calcium to one defined target above all else.

Already there appears some consistency of form and function, so why nanojunction? As indicated above, following more than 15 years of discussion on experimental data and modelling outcomes with Casey and Nick Fameli, our firm assertion is that all active nanojunctions constitute at least two adjacent biological membranes that demarcate a highly structured cytoplasmic space, with nanotubes perhaps representing the optimal operational unit in biology. To maintain junctional integrity, it is now clear that the optimal operating limit for intracellular nanojunctions is ≤ 200 nm in width, typically a few 100 nm in extension, and that either side the membrane pair must contain complementary ion transporters and channels that serve to deliver and/or receive site-specific calcium signals. As described previously, it is evident that both the ultra-structure, the electrostatic properties of each membrane of the junctional pair and the macromolecular composition of transport molecules embedded in their limiting membranes will ensure that cytoplasmic ion concentrations, and calcium in particular, are determined locally. Calcium flux may thus be specifically targeted to “receptive sites” with great accuracy, due in no small part to the ability the nanojunction to restrict diffusion of calcium away from the cytoplasmic space thus demarcated [147,172].

## 11. PM-SR Nanojunctions of Vascular Smooth Muscles

It is 40 years since analysis of electron micrographs first identified narrow cytoplasmic spaces (~20 nm across) between the PM and the superficial SR of smooth muscles [173,174]. This led to Casey van Breemen’s “superficial buffer barrier” hypothesis [145,175,176,177], which posited that restricted diffusion from this “nanospace” allows the superficial SR to limit direct calcium flux from the PM to the myofilaments [176]. This hypothesis has now received support from studies on a variety of smooth muscle types. PM-SR nanojunctions are abundant in myocytes and appear to demarcate a cytoplasmic nanodomain that coordinates the delivery of calcium to, or the removal of calcium from the SR [146,178,179,180,181,182]. In doing so these nanojunctions not only serve to regulate SR luminal calcium load, but also hyperpolarisation and relaxation, depolarisation and vasomotion, and, as we will see, may influence gene expression in parallel, in series or in isolation. PM-SR nanojunctions are therefore polymodal [71,145. They may support either vasodilation or vasoconstriction, respectively, through the capacity of the SR to not only empty when overloaded with calcium [183] or signalled to do so by vasodilators [25], but to reload its calcium store once depleted [148,184]. If we take this as an example of what is possible, then it is evident that all nanojunctions may exhibit similar levels of plasticity.

## 12. China Girl: How the Cell-Wide Web Was Weaved

In 2010 I picked up the “baton” left by Jill Clark, by reassessing her images of the SR lumen loaded with Calcium Orange, which had, we presumed, revealed a tubular network within the nucleus of living cells. Importantly, a membrane permeable marker for the nucleus had finally been made available, namely the DNA marker Draq5 (Figure 11). These further studies were also aided by the arrival of Jorge Navarro-Dorado, who joined my lab from Madrid in order to obtain a European classification for his PhD. We explored the possibility that Jill had identified cytoplasmic nanotubes within the boundary of the Draq5 labelled nucleus using Fluo-4 to report on cytoplasmic calcium. By adjusting the threshold for fluorescence detection, it was evident that a network of narrow tributaries of cytoplasm ≤500 nm wide penetrated the nucleus, perhaps reflecting the path of nuclear envelope invaginations which I had first been introduced to in 1997 whilst viewing a new confocal microscope in the Dunn School of Anatomy at Oxford [185].

I subsequently coined the name cytoplasmic nanocourses to refer to these cytoplasmic networks, which exhibited markedly higher levels of Fluo-4 fluorescence than the surrounding nucleoplasm, and could be equally well distinguished from any aspect of the wider cytoplasm, which in turn and invariably exhibited higher basal fluorescence than the nucleoplasm (Figure 12). This suggested that invaginations of the nuclear envelope might demarcate discrete signalling compartments that could be observed without the need for further image processing, irrespective of whether or not differences in fluorescence intensity resulted from differences in local cytoplasmic calcium concentration, or the influence of the local environment within each of these compartments on general Fluo-4 fluorescence characteristics [186,187]. Jorge then returned to Spain, while I finally sought direct funding to further these investigations, at the risk of revealing our findings to the “wider” research community. To my excitement and great relief, funding arrived through the award of a British Heart Foundation Programme Grant in 2012. I re-appointed Jorge as a post-doctoral researcher in 2013, who was joined a year later by a PhD student fresh from China, Jingxian Duan.

In the meantime, a new Nikon A1R+ confocal microscope had been installed in our imaging facility (IMPACT) at Edinburgh, which was critical to our further investigations due to the much lower signal-to-noise and greater spatial resolution afforded by the Galvano scanner supplied with this system (for details see [144]). Jorge and Duan spent the first 2–3 years focused on studying calcium signals within raw images of nuclear nanocourses. ER Tracker labelling and Calcium Orange loading of the SR lumen confirmed that nuclear nancourses were demarcated by invaginations of the nuclear envelope (NE) and thus the outer nuclear membrane (ONM), which is continuous with the SR (Figure 11 [72,126,188]). These studies were allied to an investigation of the distribution along the inner nuclear membrane (INM) of lamin A and histone marks. Our longer-term goal being to determine whether or not calcium signals passed from the nuclear invaginations into the nucleus to thus modulate gene expression. How wrong could we be, yet again!!

With the Nikon A1R+ set to detect those cytoplasmic nanocourses visible within the boundary of the nucleus, we frequently observed variegated, region-specific differences in Fluo-4 fluorescence intensity in the bulk cytoplasm (beyond the boundary of the nucleus). In short, highly localised, time-dependent and asynchronous fluctuations in Fluo-4 fluorescence intensity were evident across the wider cell. At first, I saw this variegated loading as an irritation that was likely due to erroneous loading of organelles, vacuoles and other aspects of the cells. However, cells presented like this far more frequently than we ever observed “uniform” loading of Fluo-4. After three years of irritation and periodic re-examination, I chose not to exclude such cells from further analysis as others had done previously (see for example [189]), rather I focused our attention on the indications derived from what could be seen in the clear majority. For when one suspended disbelief, it was all too clear that a variety of highly localised fluctuations in Fluo-4 fluorescence occurred within this variegated “map” of the cell. After viewing one particular cell during live acquisition, I requested that Duan carry out deconvolution of all Fluo-4 images within the time series acquired. She dutifully obliged; Jorge had previously stated that this was “impossible” and made no attempt to do so, which eventually settled arguments over first author status. In short, with the laser power, threshold, gain and F_max_ set to highlight nuclear nanocourses, Duan’s further image processing revealed a cell-wide network of well-defined cytoplasmic nanocourses (≤400 nm across) that appeared to be demarcated by SR, and spanned the entire cell from the Draq5 labelled nucleus to the plasmalemma. It is worth noting here that the dimensions of the nanocourses we described likely provide larger estimates than reality, because deconvolution will not have removed all stray light. Nevertheless, this was a consistent observation, irrespective of cell shape or size (Figure 13). If we consider this in the context of electron microscopy, where preparations are dehydrated and probably report smaller distances of separation of junctional membranes than is reality, it seems likely that in fully hydrated cells the true dimensions of many nanojunctions of the SR may lie somewhere between 20 and 200 nm.

During short time series’ (2–6 min; time-limited by photo-toxicity) hotspots of local calcium flux, ~200–400 nm in diameter, were readily identifiable in pseudocolour representations of this cell-wide network at rest (Figure 13), the intensity of which oscillated over a time course of seconds, without propagating beyond the nanocourse within which a given hotspot arose. Moreover, these hotspots of calcium flux exhibited asynchronous temporal characteristics when compared to adjacent hotspots within the same nanocourse, or hotspots arising in different nanocourses. It is possible, therefore, that these hotspots of calcium flux might represent activity akin to the “calcium sparklets” previously described in visceral and vascular myocytes [126,189]. Here too, a high degree of regional variability in the spatiotemporal characteristics of spontaneous SR calcium release was noted, which was proposed to be due to the presence of discrete calcium release units that are regulated autonomously [126]. However, the authors of these studies concluded that junctional complexes such as the PM-SR junction were not required for the extensive patterns of calcium release observed. To quote: “a diffusion barrier *per se* seems an unlikely explanation for some extended patterns of release which followed the surface membrane, it is more likely that such events come about as a result of the spatial distribution of RyRs near the membrane” [189]. Our studies are consistent with both proposals, but suggest yet more. They demonstrate that junctional membrane complexes, variations in RyR subtype, and variations in the spatial distribution of RyR clusters contribute to regional variations in the pattern of calcium flux from local release sites in smooth muscles. Accordingly, distances of separation between hotspots for subplasmalemmal (~350 nm) and nuclear nanocourses (350 nm) [144] are consistent with those for skeletal muscle RyR1s [166], while distances of separation for extra-perinuclear (~400 nm) and perinuclear (~450 nm) nanocourses are significantly greater [144] and closer to those reported for cardiomyocyte RyR2s (0.6–0.8 µm) [115]. This is significant, because these distances of separation are entirely consistent with the regional distribution of RyR1 and RyR2 in pulmonary arterial myocytes previously reported by my laboratory and others [103,108,137]. Consistent with this, temporal fluctuations in the fluorescence intensity of hotspots were markedly attenuated by prior depletion of SR calcium stores by SERCA inhibitors and abolished upon blocking RyRs. In short, these events most likely reflect low level, basal calcium flux (leak) from the SR via RyRs. However, while RyRs can remain open for many seconds, the fastest gating events are on the millisecond time scale [190]. Therefore, the development of confocal systems with higher temporal and spatial resolution is required before we can measure the kinetics of hotspots of calcium flux with sufficient precision to determine whether or not they are generated by “unitary” calcium release through discrete RyRs and/or RyR clusters.

Supporting the view that the wider network of cytoplasmic nanocourses may represent a circuit for cell-wide communication, it was evident that a subpopulation of LysoTracker Red labelled endolysosomes migrated through this network of cytoplasmic nanocourses, while, consistent with our previous observations [108,116], a larger, static cluster was evident in the perinuclear region of these cells [144]. By contrast, all MitoTracker Red labelled mitochondria formed static clusters in acutely isolated cells [144], as reported previously by others [191], which sit within the nanocourse network where they too likely form nanojunctions with the SR [192].

Site- and function-specific calcium signalling was confirmed using a membrane-traversing peptide from scorpion venom that selectively activates RyR1, namely maurocalcine [193]. Consistent with the distribution of RyR1s, maurocalcine preferentially increased calcium flux into subplasmalemmal and nuclear nanocourses (Figure 14 [144]). Maurocalcine also evoked concomitant myocyte relaxation, confirming our previous proposal that RyR1s might coordinate vasodilation [25,103]. By contrast there was relatively little change in calcium flux within even the most proximal extra/perinuclear nanocourses. In short, we had likely visualised for the first time unloading of SR calcium through RyR1s within the “superficial buffer barrier” presumed to be demarcated by PM-SR nanojunctions [145], which confer nanoscale path lengths and have long been predicted to coordinate calcium removal from the SR and thus relaxation [25,71,103,183], as well as SR refilling during prolonged contraction [60,148,176,184,194,195,196,197,198].

Curiously, however, over the time course of our experiments (2–6 min) maurocalcine-induced myocyte relaxation was not accompanied by concomitant falls in calcium flux into extra/perinuclear nanocourses. If anything, asynchronous calcium flux continued within these nanocourses, with perhaps slight increases in activity but no evidence of cell-wide signal propagation. One explanation for this could be that the relatively small population of RyR1s in extra/perinuclear nanocourses neither face nor couple with the contractile apparatus, rather they act to direct calcium flux towards PM-SR nanojunctions via SERCA2b and away from SR release sites occupied by RyR2s/RyR3s that guide myofilament contraction.

As one might expect of a vasoconstrictor, angiotensin II induced a calcium wave that propagated throughout all extraperinuclear and perinuclear nanocourses, but not subplasmalemmal nanocourses, and triggered concomitant myocyte contraction (Figure 14). Surprisingly, however, this propagating signal was immediately preceded by a rapid fall in Fluo-4 fluorescence intensity within the majority of cytoplasmic nanocourses, except for those at the point of wave initiation. This suggests that angiotensin II might also act to pre-load the SR with calcium, which could be a critical step prior to induction of cell-wide signal propagation and myocyte contraction, and may, for example, rely heavily on RyR sensitisation through increases in calcium-calsequestrin binding within the lumen of the SR [199,200,201]. Importantly, angiotensin II-induced calcium signals were blocked by, you guessed it, 8-bromo-cADPR. Given that cADPR preferentially activates RyR1s and RyR3s [109] but can only sensitise RyR2s to CICR [110], it therefore seems likely that angiotensin II-evoked cADPR accumulation within extraperinuclear nanocourses may serve to activate local subpopulations of RyR1s and/or RyR3s [109] while delivering concomitant sensitisation of RyR2s to CICR [110]. In this way subsequent initiation of a propagating calcium signal and thus myocyte contraction could be permitted. Accordingly, prior depletion of SR stores with thapsigargin and block of RyRs with tetracaine abolished angiotensin II-evoked calcium signals [144].

If, however, we consider the extra-perinuclear location of the site of signal initiation in response angiotensin II (Figure 14), this rather surprisingly favours RyR1s over RyR3s, because RyR3s are so heavily restricted to the perinuclear SR. If this is the case, then the action of maurocalcine (see above) lends support to the view that a critical factor in the selection of vasoconstriction over vasodilation might be the capacity of a vasoconstrictor to pre-load the SR prior to RyR activation [113,199,200,202,203]. In this respect it is worth noting that the studies of others have suggested that cADPR might act to accelerate SR-filling by activating SERCA [204,205,206], although our studies have revealed no clear evidence of this [109]. Such an action could provide an alternative explanation for the all-or-none block of HPV by 8-bromo-cADPR, if 8-bromo-cADPR blocks the initiation of HPV by inhibiting the activity of both SERCA and RyR1, 2 and 3.

It is important to note that although the widths of all extra/perinuclear nanocourses are ≤400 nm across, when considering contraction the larger scale of nanocourses is not entirely incompatible with limits imposed by models of calcium flux across nanojunctions, or electron micrographs. As mentioned previously, calcium diffusion and signal propagation can be further limited by RyR binding partners, the organisation of the SR, and the presence or absence of structures that span the two limiting membranes of any given nanojunction (for review see [113,114]). It is therefore notable that separate PM regions have been described for filament attachment and caveolae [207], while the density of myosin filaments appears to be less in the cell periphery than in the central myoplasm [208]. In addition, the functional calcium-binding protein calmodulin is tethered proximal to the SR membranes that line myofilament arrays [209], rather than being freely diffusible in the cytoplasm. All relevant path lengths from the SR to calcium binding proteins, and from calcium binding protein to myofilaments may therefore be on the nanoscale, even if the distances of separation for the junctional membranes are greater than 200 nm.

At this point it is worth highlighting a further, long-standing curiosity that remains to be resolved. Our studies on pulmonary arterial myocytes and those of others have demonstrated that IP_3_Rs do not couple by CICR to RyRs in this cell type [67,117], which is contrary to the findings of studies on, for example, venous smooth muscles [210,211]. This strongly suggests that RyRs and IP_3_Rs might be segregated in pulmonary arterial myocytes, perhaps in part by SR nanojunctions that demarcate discrete cytoplasmic nanocourses.

## 13. Nuclear Invaginations: A Road for El Dorado

As mentioned above, maurocalcine also increased calcium flux into nuclear nanocourses adjacent to relatively inactive perinuclear nanocourses. This further exposed the functional segregation of nuclear nanocourses from their nearest neighbour, through the strategic targeting of RyR1s to the outer nuclear membrane (ONM) that demarcates nuclear nanocourses. Against all expectations, however, maurocalcine-evoked calcium flux within nuclear nanocourses did not propagate freely into the nucleoplasm to any great extent, i.e., calcium is released across the ONM into the cytoplasmic nanocourses demarcated by each active invagination, but neither directly nor indirectly into the nucleoplasm. Consistent with this, we found no evidence in quiescent cells of hotspots of calcium flux propagating into the adjacent nucleoplasm. Closer inspection of calcium flux within nuclear nanocourses also revealed functional signal segregation in response to not only maurocalcine but the vasoconstrictor angiotensin II (Figure 14). Both stimuli triggered increases in calcium flux within a subset of nuclear nanocourses, and with distinct spatiotemporal signatures evident in each of these activated nanocourses. Intriguingly, with respect to angiontensin II alone we did observe evidence of very low-level propagation of calcium flux from nuclear nanocourses into adjacent, Draq5-positive nucleoplasm [144]. Therefore, it is possible that certain physiological stimuli might gate RyRs and pathways for trans-NE calcium flux into the nucleus, while others may not. This could offer a powerful mechanism for differential regulation of gene expression by different stimuli.

The functional reasons for the isolation of nuclear nanocourses are not clear, but it may be to, for example, prevent wide-scale gene activation/inactivation events that could switch cells from a differentiated to proliferative phenotype, operated through specific changes in calcium flux. Using electron microscopy we observed 20–200 nm diameter invaginations, as have others [185], within the nucleus of arterial myocytes in-situ in arterial sections. We could distinguish invaginations of the outer nuclear membrane (ONM), forming open transnuclear channels, or shallow, blind invaginations of variable depths reaching into the nucleus (Figure 11). As the NE is a double membrane, invaginations also contained inner nuclear membranes (INM), with the luminal space between INM and ONM ranging from 10–50 nm. Jorge Navaro-Dorado labelled fixed cells for lamin A, which generally lines the INM, and this too revealed tubular networks that criss-crossed the nucleoplasm of these differentiated cells (Figure 15), in much the same way as nuclear nancourses.

Jorge then developed, later in association with Duan, novel and exhaustive image analysis protocols, by which we began to examine the possibility that calcium flux across the ONM could in some way modulate gene expression, with insightful comment and support from Eric Schirmer. Normal ovoid nuclei tend to have histones carrying both H3K9me2 and H3K9me3 marks, and the chromatin cross-linking protein barrier to autointegration factor (BAF) associating with NE proteins such as emerin and making the nuclear periphery generally silencing [212,213,214]. Interestingly, these marks segregate in differentiated arterial myocytes with the H3K9me2/3 both still at the outer limits of the nucleus but depleted with respect to BAF, and the nuclear invaginations rich with H3K9me2 and BAF (Figure 15), but depleted with respect to H3K9me3 [144]. The combination of H3K9me2/3 together is strongly silencing, but absent the me3 mark and the me2 can reflect a poised state that has been found at myogenic regulators such as the myogenin promoter [215]. It is therefore possible that the non-propagating calcium transients in distinct invaginations in some way specifically regulate chromatin in differentiated cells, as the different chromatin marks are concentrated in puncta. Discrete H3K9me2-lamin A puncta (≈ 500 nm diameter) were separated by ≈ 400 nm, while emerin-BAF puncta (≈ 400 nm in diameter) were separated by ≈ 500 nm, approximating the 350 nm spacing between the tetracaine-sensitive hotspots of nuclear nanocourses. Potentially calcium and/or charge responsive and functionally distinct chromatin domains may therefore be established by nuclear invaginations. Supporting this, blocking RyRs with tetracaine reduced the expression of two genes of interest, one encoding the DNA mismatch repair protein MutL homolog 1 (*Mlh1*), which can be repressed through interaction with H3K9me2 [216,217], and another encoding the S100 calcium binding protein A9 (*S100a9*), which can be repressed by BAF [218].

Therefore, consistent with predictions based in large part on the actions of 8-bromo-cADPR against pulmonary artery dilation and constriction, respectively, it is now evident that multiple coordinated actions may very well be delivered by distinct nanojunctions of the SR that demarcate diverse nanocourse networks, to thus provide for signal segregation sufficient to enable nanocourse-specific delivery of calcium signals with the capacity to coordinate the full panoply of cellular processes.

## 14. More Than This: Lost in Proliferation

During the transition to proliferating myocytes in culture, the entire, cell-wide network of cytoplasmic nanocourses was lost, inclusive of all lamin and emerin positive nuclear invaginations [144]. These observations are therefore consistent with the idea that invaginations act to regulate anti-proliferative genes, that is until the proliferative phenotype [219] is ready to be engaged. During proliferation we noted loss of S100A9 expression (but not MLH1 expression), namely the emerin and BAF regulated gene whose expression was attenuated by blocking calcium flux through RyRs with tetracaine, which is consistent with previous reports on S100A9 repression during proliferation of airway smooth muscles [220]. These observations, the distribution of chromatin marks and general tendency of NE-association to keep chromatin repressed [221,222] lends support to the view that NE invaginations may play a role in genome regulation and cycles of gene repression and activation. That this phenotypic change is delivered through reconfiguration of the cell-wide nanocourse network that directs calcium flux is further highlighted by:

A switch in dependency of angiotensin II-induced calcium transients from RyRs to IP_3_Rs;

Unrestricted, cell-wide SR Ca^2+^ release due to loss of cytoplasmic nanocourses;

Loss of the “nuclear buffer barrier” [223] that opposed direct calcium flux into the nucleoplasm in acutely isolated cells.

Accordingly, others have found that myocyte proliferation coincides with whole-scale changes in gene expression inclusive of decreases in lamin A [224] and RyR expression, and augmented IP_3_R expression [225].

## 15. Reviewing the Situation: Got to Pick a Pocket or Two Boys

From a curious piece of pharmacology that was the all-or-none block of HPV by 8-bromo-cADPR, and the cautious proposal that outcomes may be explained by the presence of an intracellular junction [24], twenty years on we reveal the cell-wide web (Figure 16).

This cell-wide network of cytoplasmic nanocourses appears to be delineated by membrane-membrane nanojunctions of the SR, which provide discrete lines of communication that span the entire cell. Functional specification may well, therefore, be determined by the constraints on calcium diffusion imposed by SR junctional membranes, as predicted by computer models [147], and by the strategic positioning of different types of calcium transporters and release channels targeted to them, through unique kinetics, affinities for calcium and mechanisms of regulation [11]. The structural elements that hold this junctional network in place remain to be determined, but there is evidence to suggest that junctophilins may be critical to the formation of PM-SR nanojunctions at the very least [226,227,228,229], while others suggest, on the basis of the action of depolymerising agents (e.g., nocodazole), that actin, tubulin and/or other components of microtubules may provide further structural support [230].

In pulmonary arterial myocytes calcium flux through RyR1s located within PM-SR nanocourses evokes relaxation with no evidence of cell-wide signal propagation, confirming the existence of a functional “superficial buffer barrier” [145]. Distinct nanocourses rich in RyR2s and RyR3s carry rapid, propagating calcium signals that cross the entire cell from pole to pole triggering myocyte contraction. Yet these latter signals did not enter those nanocourses demarcated by PM-SR nanojunctions, which constitute the superficial buffer barrier. Therefore, RyR2 subtype designation determines the capacity for rapid signal propagation by CICR across thus specified nanocourses, as it does so effectively in cardiac muscle [113].

On one hand this allows cell-wide synchronous actions as required. On the other it may additionally confer the capacity for onward transfer (perhaps sampling) of a fraction of released calcium through SERCA that feed functionally distinct subsets of nanocourses. If we consider nuclear invaginations in this respect, such sampling of released calcium may serve to inform the nucleus on activity-dependent requirements of the cell. Invaginations of the nuclear envelope confer further segregated and diverse networks of cytoplasmic nanocourses that project deep into the nucleoplasm, which break down themselves into multiple subtypes based on emerin-BAF and lamin A-H3K9me2 clusters, are served by a unique pairing of ONM resident calcium pumps (SERCA1) and release channels (RyR1) and carry calcium signals with different spatiotemporal characteristics when activated. Here, regulated calcium flux across the ONM into nuclear nanocourses might contribute additional levels of genome regulation by, for example, segregating specific chromatin types for cycles of reactivation in differentiated cells and providing a path for related gene repression during phenotypic modulation [219]. Importantly, others have confirmed the regional distribution of SERCA and RyRs not only in pulmonary arterial smooth muscles [137], but in airway smooth muscles [138], which may provide for the high degree of variability in kinetics of spontaneous SR calcium release in myocytes from the ileum [189] and hepatic portal vein [126]. It is also notable that previous studies have identified RyR1, RyR2 and RyR3 expression in cerebral arterial myocytes [139], which must at some stage inform current models on calcium signalling in these cells too, which for some reason rely solely on a consideration of RyR2 clusters, whether one considers PM-SR nanojunctions [230,231] or more recent “comparative” assessment of the role of lysosome-SR nanojunctions in this cell type [129].

Importantly, and regardless of the functional subdivision of cytoplasmic nanocourses, all path lengths from calcium release site to targeted signalling complex appear to be of the order of tens or hundreds of nanometres, with picolitre volumes of cytoplasm lying within the boundaries of each nanocourse [147]. Relatively small net increases in local calcium flux (1–2 ions per picolitre) will therefore be sufficient to raise the local concentration into the affinity ranges of most cytoplasmic calcium binding proteins [147]. Thus, calcium binding proteins may act much like local “switches”, operated through changes in local, perhaps unitary calcium flux that coordinate nanocourse-specific functions by adjusting the relative probability of a given switch sitting at some conformational state between *OFF* and *ON*. Moreover, in this way coincident increases in calcium flux could be triggered in two distant parts of the cell at the same time, to coordinate, for example, myocyte relaxation and associated gene expression regulation. This draws obvious parallels to mechanisms of conduction in single-walled carbon nanotubes, which behave as quantum wires that transmit charge carriers through discrete conduction channels, enabling memory, logic and parallel processing. Thus, by analogy, our observations point to the incredible signalling potential that may be afforded by modulating quantum calcium flux on the nanoscale, in support of network activities within cells with the capacity to permit stimulus-dependent orchestration of the full panoply of diverse cellular processes. Perhaps more importantly, this cellular intranet and its associated network activities are not hardwired, reconfiguring to deliver different outputs during phenotypic modulation on the path, for example, to cell proliferation. In a similar way the cell-wide web may provide the capacity for cell- and system-specific outputs, because nuclear envelope invaginations are a feature of many cell types [15,16,17,18,19], indicating that the cell-wide web and its constituent nuclear nanocourses may vary in nature between different cell types in order to meet the functional requirements of any given cell [11,149,188,232].

In conclusion, it is time for the field of cellular signalling to move away from casual reference to microdomains, nanodomains and nanospaces, because these are all meaningless adjustments in terminology in the context of cellular signalling. They simply indicate smaller size or increased resolution, and offer no true functional insight, with the possible exception of organelle-specific autoregulation. Moreover, given that estimates of the volume of cytoplasm demarcated by SR nanojunctions are of the order of picolitres [147,172], when considering cytoplasmic domains or spaces they are all inaccurate terms. In short, it is the dimensions of the nanojunction or nanotube that are critical to effective control of site-specific ion flux or exchange across the cell-wide web. So, when it comes to appropriate and informed use of terminology, nanojunction it is.

## Figures and Tables

**Figure 1 molecules-25-04768-f001:**
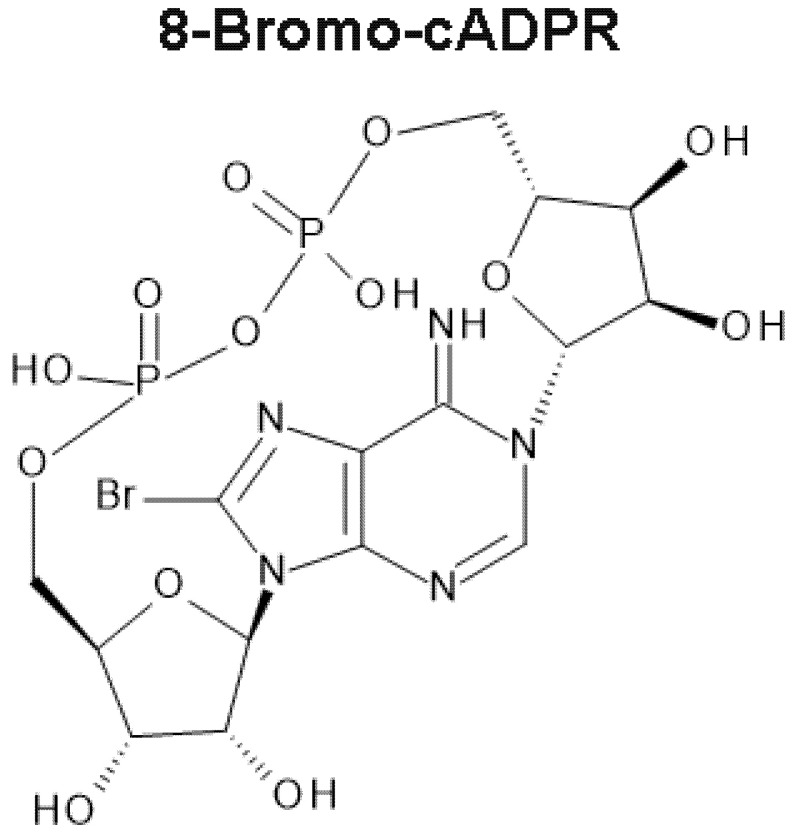
8-Bromo-cyclic ADP-ribose.

**Figure 2 molecules-25-04768-f002:**
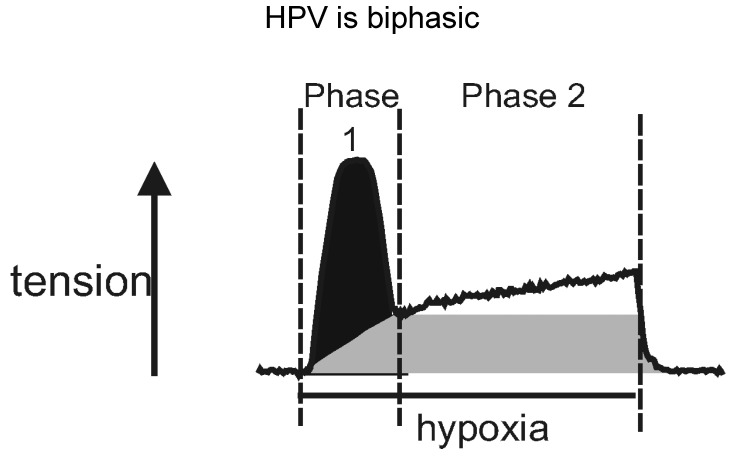
Schematic diagram that describes the two phases of hypoxic pulmonary vasoconstriction (HPV) triggered upon exposure of an isolated pulmonary artery ring to hypoxia. Phase 1 is driven by the smooth muscle (black). Phase 2 is driven by the smooth muscle (grey) and amplified through an endothelium-dependent mechanism (white).

**Figure 3 molecules-25-04768-f003:**
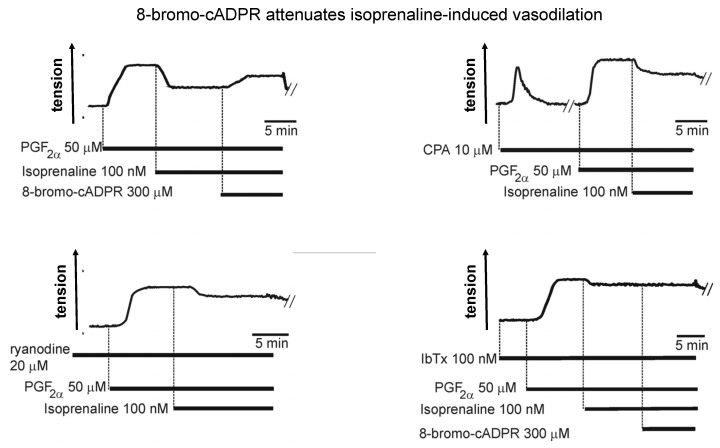
Example records show that the cADPR antagonist 8-bromo-cADPR (**top left**), ryanodine receptor block with ryanodine (**bottom left**) and depletion of sarcoplasmic reticulum calcium stores with cyclopiazonic acid (CPA; **top right**) attenuates (~60%) pulmonary artery dilation induced by Isoprenaline. Blocking BK_Ca_ channels with iberiotoxin delivers ~90% attenuation (**bottom right**).

**Figure 4 molecules-25-04768-f004:**
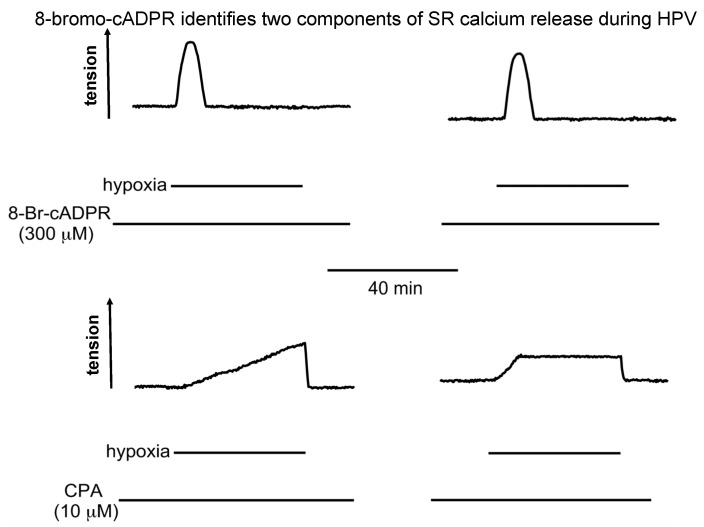
Example records show that pre-incubation of pulmonary artery rings with 8-bromo-cADPR, a cADPR antagonist, blocked phase 2, but not phase 1, of HPV (**upper panels**). By contrast, depletion of sarcoplasmic reticulum calcium stores with cyclopiazonic acid (CPA) blocked phase 1, but not phase 2, of HPV (**lower panels**).

**Figure 5 molecules-25-04768-f005:**
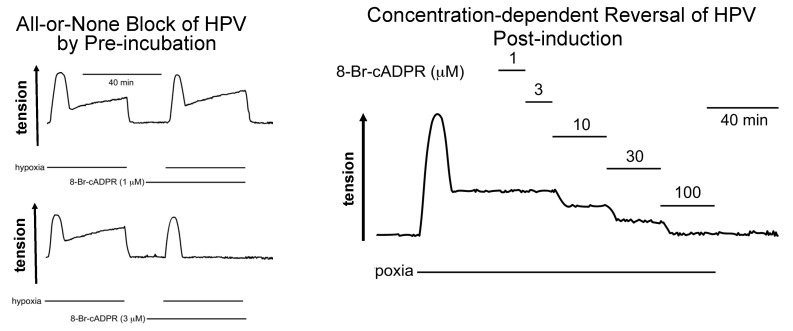
Pre-incubation of pulmonary artery rings with 8-bromo-cADPR blocks hypoxic pulmonary vasoconstriction (HPV) in an all-or-none manner (**left** hand panels). By contrast, when applied after prior induction of HPV in pulmonary arteries without endothelium, 8-bromo-cADPR reverses HPV in a concentration-dependent manner (**right** hand panel).

**Figure 6 molecules-25-04768-f006:**
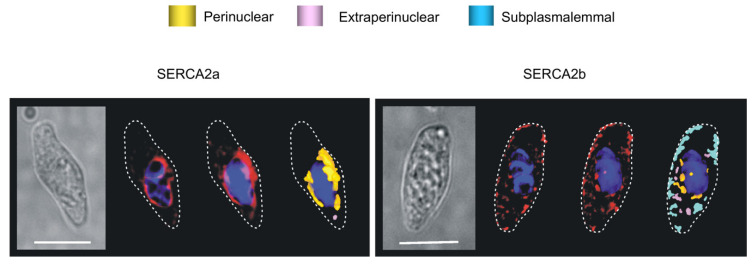
Example cells showing labelling for SERCA2a (**left** hand series of panels) and SERCA2b (**right** hand series of panels) in pulmonary arterial myocytes. In each case, from left to right, a bright field image, z section, 3D reconstruction and 3D digital representation of labelling colour coded by region of the cell are shown. Nucleus identified by DAPI labelling (navy blue). Scale bar, 10 μm.

**Figure 7 molecules-25-04768-f007:**
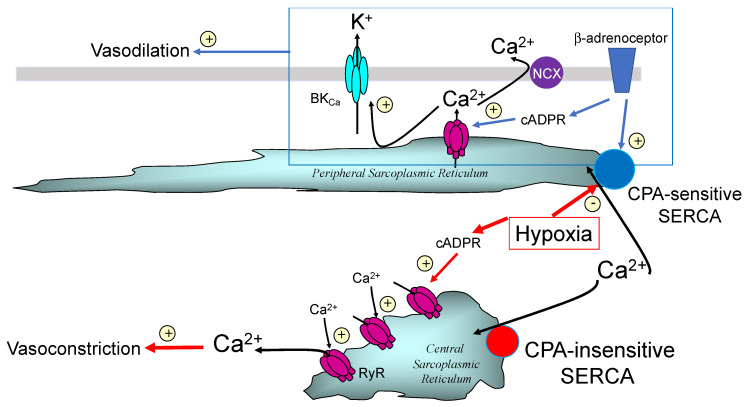
Schematic representation of an early two compartment model developed to explain the curious pharmacology of 8-bromo-cADPR and cyclopiazonic acid.

**Figure 8 molecules-25-04768-f008:**
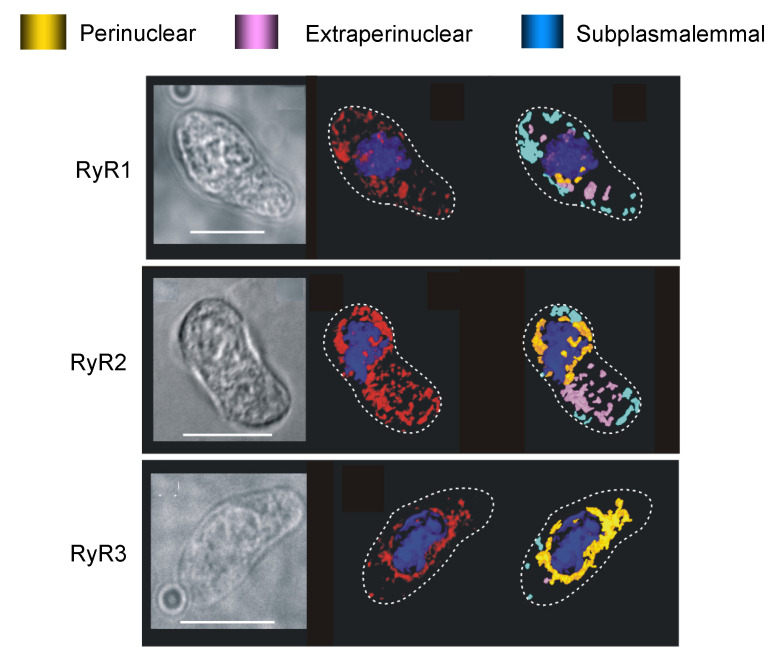
Example cells showing labelling for RyR1 (**upper panels**), RyR2 (**middle panels**) and RyR3 (**lower panels**) in pulmonary arterial myocytes. In each case, from left to right, a bright field image, 3D reconstruction and 3D digital representation of labelling colour coded by region of the cell are shown. Nucleus identified by DAPI labelling (navy blue). Scale bars, 10 μm.

**Figure 9 molecules-25-04768-f009:**
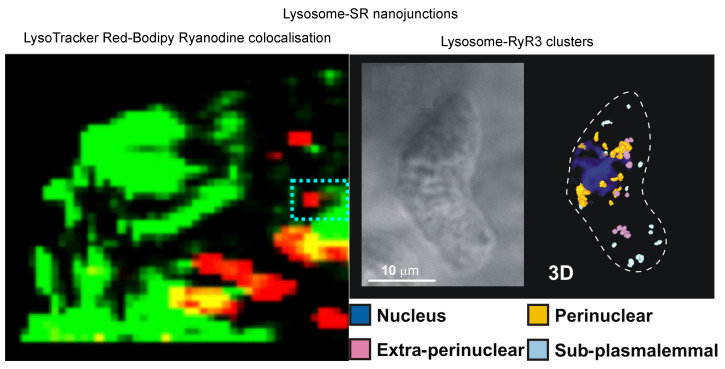
Example images showing lysosome-SR nanojunctions. Left hand image shows enlarged segment of an acutely isolated pulmonary arterial myocyte, that identifies nanojunctions (yellow) between LysoTracker Red labelled acidic stores and Bodipy-ryanodine positive SR in an acutely isolated pulmonary arterial myocyte. Dashed blue rectangle (vertical axis, 0.5 μm) identifies a Lysotraker Red labelled organelle and Bodipy-ryanodine labelled SR in the same focal plane, but separated by ~0.5 μm. Right hand panels show a brightfield image (**left**) and a 3D reconstruction (**right**) of a fixed cell labelled for lysosomes (αIGP120), RyR3 and the nucleus (DAPI, navy blue), that identifies lysosome-RyR3 clusters colour coded by region of the cell.

**Figure 10 molecules-25-04768-f010:**
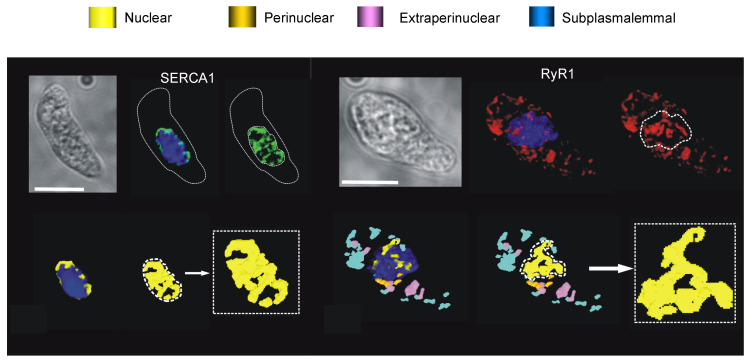
Example cells showing labelling for SERCA1 (**left** hand panels) and RyR1 (**right** hand panels) in pulmonary arterial myocytes. In each case, upper panels show, from left to right, a bright field image, and 3D reconstructions with and without DAPI labelling (navy blue = nucleus). Lower panels show 3D digital representation of labelling colour coded by region of the cell. Scale bars, 10 μm.

**Figure 11 molecules-25-04768-f011:**
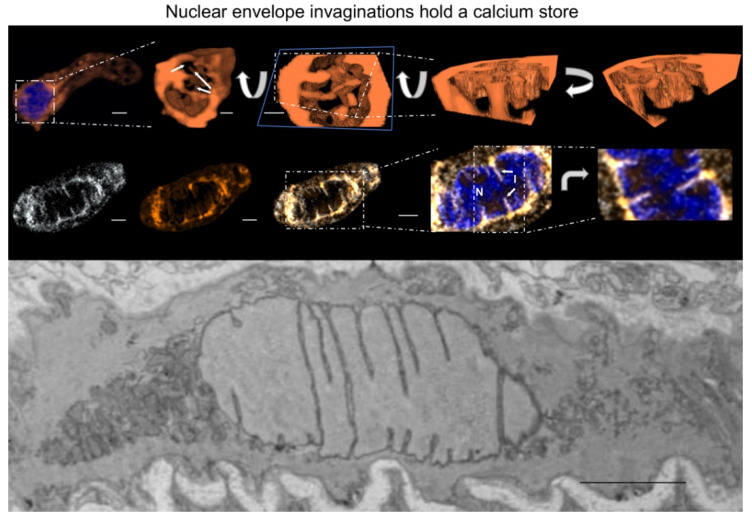
Upper panels show (from left to right) nuclear invaginations at differing orders of magnification, in sections through the nucleus of a pulmonary arterial myocyte in which the lumen of the sarcoplasmic reticulum has been loaded with Calcium Orange (Scale bars from left to right, 2 μm, 1 μm, 1 μm). Middle panels show ER tracker and Calcium Orange co-labelelling of nuclear invaginations. Draq5 identifies the nucleus (blue) in each case (Scale bars, 2 μm). Lower panel, electron micrograph shows nuclear invaginations in a pulmonary arterial myocyte in-situ in a pulmonary artery section (Scale bar, 2 μm).

**Figure 12 molecules-25-04768-f012:**
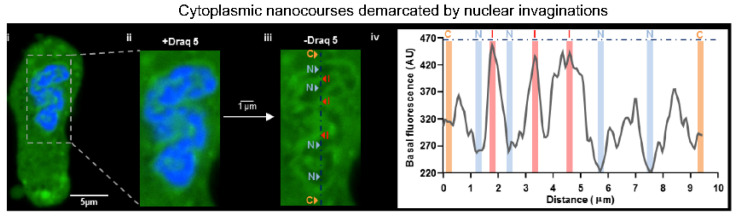
Fluo-4 fluorescence (green) identifies cytoplasmic nanocourses within the boundary of the Draq5 labelled nucleus (blue) of a confocal Z section through a pulmonary arterial myocyte (**left** hand panels i-iii). A spectral intensity plot across the nucleus (**right** hand panel iv) shows that nanocourses within nuclear invaginations (NI) exhibit higher levels of fluorescence than the nucleoplasm (N), and cytoplasm (C) identified by the dashed line in panel iii.

**Figure 13 molecules-25-04768-f013:**
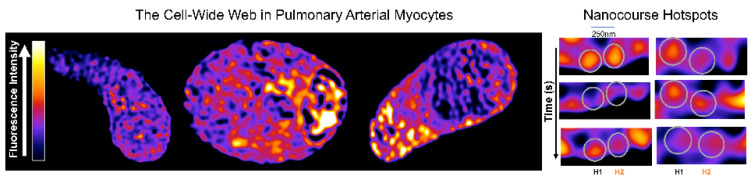
From left to right, panels show pseudocolour representations of deconvolved z sections of Fluo-4 fluorescence that identify a cell-wide network of cytoplasmic nanocourses within three different acutely isolated pulmonary arterial myocytes, irrespective of cell size or shape. Right hand panels show enlarged images of two cytoplasmic nanocourses from the same cell, in which asynchronous, time-dependent fluctuations in hotspots of calcium flux (from top to bottom) can be seen.

**Figure 14 molecules-25-04768-f014:**
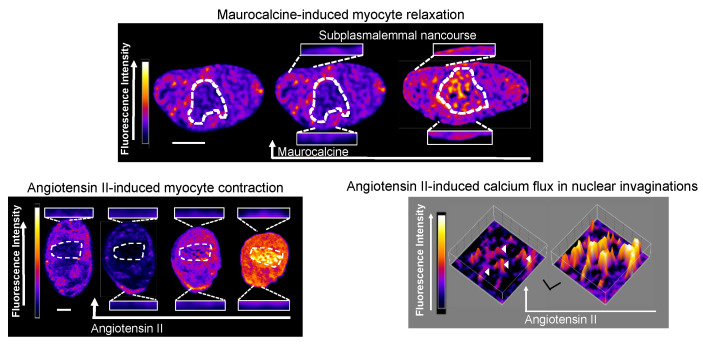
Upper panels, from left to right, show myocyte relaxation in response to RyR1 activation by maurocalcine, with nanocourse-specific changes in calcium flux identified in pseudocolour representations of deconvolved z sections of Fluo-4 fluorescence (scale bar, 2 μm). Lower left, panels similarly show calcium flux during myocyte contraction induced by angiotensin II (scale bar, 3 μm). Lower right, panels show angiotensin II-induced increases in calcium flux within nuclear nanocourses of a different cell (scale bar, 1 μm).

**Figure 15 molecules-25-04768-f015:**
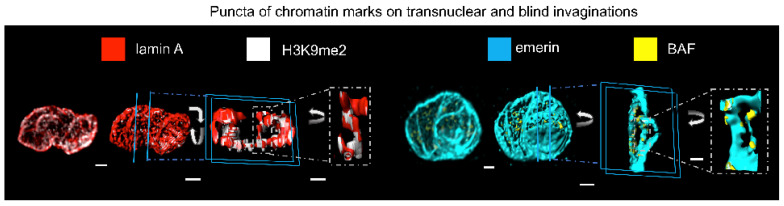
Right hand panels, sections through a 3D reconstruction of the nucleus of a pulmonary arterial myocyte reveal puncta of lamin A and H3K9me2 colocalisation on a lamin A positive transnuclear invagination (from left to right scale bars 1 μm, 1 μm, 500 nm). Left hand panels, sections similarly show puncta of emerin and BAF (barrier to autointegration factor) colocalisation on transnuclear and blind invaginations (from left to right scale bars 1 μm, 1 μm, 500 nm).

**Figure 16 molecules-25-04768-f016:**
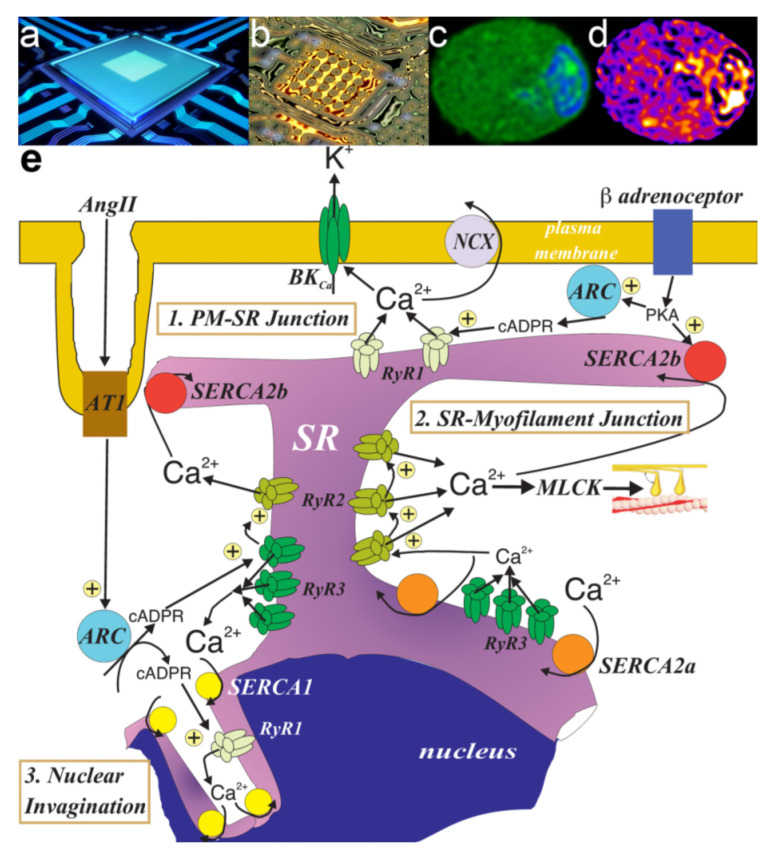
Calcium signalling is analogous to quantum tunneling across a cell-wide circuit with the nucleus at its centre. (**a**) and (**b**), Microprocessor at the centre of a circuit board. (**c**) (Fluo-4, green) and (**d**) (Fluo-4, pseudocolour), show the nucleus (Draq5 in (c) only, blue) at the centre of a cell-wide circuit of cytoplasmic nanocourses. e, Schematic shows calcium flux across cytoplasmic nanocourses demarcated by junctions between the plasma membrane and sarcoplasmic reticulum (PM-SR nanojunction), the sarcoplasmic reticulum nanojunctions aligned with the contractile myofilaments of a smooth muscle and the nuclear invaginations. Angiotensin II, AngII; Ryanodine receptor, RyR; Sarco/endoplasmic reticulum calcium ATPase, SERCA; sodium/calcium exchanger, NCX; ADP-ribosyl cyclase, ARC; Myosin light chain kinase, MLCK; AT1, Angiotensin AT1 receptor; Protein kinase A PKA; large conductance calcium-activated potassium channel, BK_Ca_. Panels (a) and (b) are free images. The schematic in panel (**e**) was adapted from previous versions developed and published by AME. Reprinted from Publication title, Vol /edition number, Author(s), Title of article / title of chapter, Pages No., Advances in Pharmacology, 78, Evans AM, Nanojunctions of the Sarcoplasmic Reticulum Deliver Site- and Function-Specific Calcium Signaling in Vascular Smooth Muscles, 1–47 Copyright (2017), with permission from Elsevier.

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
