# Peer review of "On a Magical Mystery Tour with 8-Bromo-Cyclic ADP-Ribose: From All-or-None Block to Nanojunctions and the Cell-Wide Web"

_molecules, 2020, doi:10.3390/molecules25204768_

Round 1

Reviewer 1 Report

It is a rather unusual way of presenting the text. The review is set up as a first-person telling that the author makes about the story of his research carried over the years. He defines the  sequence of findings during his scientific work, integrated with results from literature up to descovery of cADPR, its antagonist and NAADP,  with a clear and critical evaluation of the results gradually obtained. Far from being easy, the topic is discussed simply and the author is able to develope clearly the story from the beginning of the research to the final results, highlighting the contributions of his co-workers and people outside his team. It is a critical analysis of all research step, improved each time by the new discoveries, and interconnecting Physiology with Pharmacology and Biochemistry (namely the basic studies on NAD-derived compounds since the nineties). It is a good and widely useful exercise to retrace the salient steps of a scientific career leading to the final proposed new models. References are updated. It is acceptable that  numerous articles are self-referenced, being the author among the main researchers of the proposed topic

English is of good quality. Text requires minor revisions for typing mistakes, as:

Line 5, A. Mark A. Mark Evans  > A. Mark Evans

Line 155,  teh>the

Manuscript is acceptable as it is.

Reviewer 2 Report

Summary:

This is an outstanding and very clearly written review by Mark A Evans regarding intracellular Ca2+ signaling mechanisms mediating hypoxic and drug induced vasoconstriction and vasodilation in pulomonary arterial myocytes and vascular rings. The experimental findings over the past 30 years  review are described in exquisite detail also provides insight into the personal interrelationships and thinking that underlie the described findings. We learn about their discussions that led to clarify for how the same type of hypoxic induced Ca2+ transient induces either vasoconstriction or vasodilation in the aforementioned tissues.  The development of the current model had its origin based on complex effects of 8-bromo cADPR, a selective antagonist of ryanodine receptor-activated calcium release from intracellular calcium stores, in these tissues.  We are told about how the development of novel technology and a broader   selective armamentarium of selective reagents were critical in resolving details underlying the current model. These advances help account for how Ca2+ transients induce either contraction or relaxation of vascular smooth muscle. The model envisions segregated signaling domains made up of membrane delimited nano-junctions that possess different arrays of transporters and receptors within the sarcoplasmic reticulum of vascular smooth muscle. Even though the transporters mediate ATP-dependent sarcoplasmic reticular active Ca2+uptake, they may induce different responses to hypoxia since their spatial distribution  and drug sensitivity is not the same in different regions of the endoplasmic reticulum. Another kind of heterogeneity between different regions of the sarcoplasmic reticulum within these nano-junctional microdomains of perhaps 200 nm separating closely apposed membranes includes variability in ryanodine receptor isoform expression patterns.  Another recent very amazing finding that helps us better understand how a hypoxic or drug-induced Ca2+ transient mediates opposing responses stems from the identification by the authors’ team of cytoplasmic nanocourses, which are tubular networks extending from the cell nucleus out to the plasmalemma. In addition to this clear presentation of the evolution of this well-documented model, the author also openly describes some confounding results that still require future clarification.

Recommendation: This review is a highly informative and exhaustive review that is a pleasure to read.  It should be at the top of any ones reading list if they are searching for a reference describing how receptor and ambient induced Ca2+ signaling induces opposing responses in tissues controlling vascular perfusion.